# Approximate Decomposable Submodular Function Minimization for Cardinality-Based Components

**Nate Veldt**[*]
nveldt@tamu.edu
Texas A&M University

**Austin R. Benson**
arb@cs.cornell.edu
Cornell University

**Jon Kleinberg**
kleinberg@cornell.edu
Cornell University

## Abstract

Minimizing a sum of simple submodular functions of limited support is a special case of general submodular function minimization that has seen numerous applications in machine learning. We develop fast techniques for instances where components in the sum are cardinality-based, meaning they depend only on the size of the input set. This variant is one of the most widely applied in practice, encompassing, e.g., common energy functions arising in image segmentation and recent generalized hypergraph cut functions. We develop the first approximation algorithms for this problem, where the approximations can be quickly computed via reduction to a sparse graph cut problem, with graph sparsity controlled by the desired approximation factor. Our method relies on a new connection between sparse graph reduction techniques and piecewise linear approximations to concave functions. Our sparse reduction technique leads to significant improvements in theoretical runtimes, as well as substantial practical gains in problems ranging from benchmark image segmentation tasks to hypergraph clustering problems.

## 1  Introduction

Given a ground set $V$, a function $f \colon 2^V \to \mathbb{R}$ is submodular if for every $A, B \subseteq V$ it satisfies $f(A) + f(B) \geq f(A \cap B) + f(A \cup B)$. Submodular functions are ubiquitous in combinatorial optimization and machine learning, arising, e.g., in image segmentation [22], hypergraph clustering [30], data subset selection [44], document summarization [31], and dense subgraph discovery [43]. Algorithms for minimizing submodular functions are well studied. Strongly-polynomial time algorithms for general submodular minimization exist [36, 17, 18], but their runtimes are impractical in most cases. The past decade has witnessed several advances in faster algorithms for *decomposable submodular function minimization* (DSFM), i.e., minimizing a sum of simpler submodular functions [9, 24, 39, 29, 35, 10, 21, 19]. This problem is defined by identifying a set $E \subseteq 2^V$ of subsets of the ground set. The goal is then to solve

$$\text{minimize}_{S \subseteq V} \; f(S) = \sum_{e \in E} f_e(S \cap e), \tag{1}$$

where for each $e \in E$, $f_e$ is a submodular function supported on a subset $e \subseteq V$. Functions of this form often arise as energy functions in computer vision [22, 23, 13], and generalized cut functions for hypergraph clustering and learning [12, 28, 30, 32, 41, 42], among other applications.

This paper focuses on *cardinality-based* DSFM (Card-DSFM), where every component $f_e$ in the sum is a concave cardinality function, i.e., $f_e(A) = g_e(|A|)$ for some concave function $g_e$. A single concave cardinality function is effectively a function of one variable and is trivial to minimize. However, set functions obtained via sums of these functions are much more complex and have broad modeling power, making this one of the most widely studied and applied variants since the earliest

---

[*]This research was performed while the author was a postdoctoral associate at Cornell University.

35th Conference on Neural Information Processing Systems (NeurIPS 2021).

work on DSFM [24, 39]. In terms of theory, previous research has addressed specialized runtimes and solution techniques [24, 39, 22, 41]. In practice, cardinality-based decomposable submodular functions frequently arise as higher-order energy functions in computer vision [22] and set cover functions [39]. The most widely used generalized hypergraph cut functions are also cardinality based [30, 32, 41, 42, 1]. Even previous research on algorithms for the more general DSFM problem tends to focus on cardinality-based examples in experimental results [9, 39, 19, 29].

We present the first approximation algorithms for Card-DSFM, using a purely combinatorial approach that relies on approximately reducing (1) to a *sparse* graph cut problem. The fact that concave cardinality functions are representable by graph cuts has previously been noted in different contexts [22, 39, 41]—this can be accomplished by combining simple graph *gadgets*, whose cut properties together model the function. However, previous techniques are limited in that they (i) focus only on exact reduction techniques, (ii) do not consider the density of the resulting graph, and (iii) do not address any questions regarding the optimality of such a reduction. Here we develop new approximate reduction methods leading to a sparse graph, which we show is optimally sparse in terms of the standard gadget reduction strategy. We show that representing a concave cardinality function with a sparse graph is equivalent to approximating a concave function with a piecewise linear curve, where the number of linear pieces determines the sparsity of the reduced graph. We develop a new algorithm for the resulting piecewise linear approximation problem, and prove new bounds on the number of edges needed to model different concave functions that arise in practice. Combining our reduction strategy with state of the art algorithms for maximum flow [15, 26, 40, 14] leads to fast runtime guarantees for finding approximate solutions to Card-DSFM. Our algorithm is also easy to implement and achieves substantial practical improvements on benchmark image segmentation experiments [19, 9, 29] and algorithms for hypergraph clustering [42].

## 2 Background on Graph Reductions

A nonnegative set function $f \colon 2^V \to \mathbb{R}^+$ on a ground set $V$ is *graph reducible* if there exists a directed graph $G = (V_f, E_f)$ on a larger node set $V_f = V \cup \mathcal{A} \cup \{s, t\}$, which includes an auxiliary node set $\mathcal{A}$ as well as source and sink nodes $s$ and $t$, whose $s$-$t$ cut structure models $f$. More precisely, $f$ is graph reducible if for every $S \subseteq V$ we have

$$f(S) = \underset{T \subseteq \mathcal{A}}{\text{minimize}} \ \ \mathbf{cut}_G(\{s\} \cup S \cup T), \tag{2}$$

where $\mathbf{cut}_G$ is the cut function on the graph $G$. In words, this says that evaluating $f(S)$ is equivalent to finding the minimum $s$-$t$ cut penalty that could be obtained by placing $S$ on the $s$-side of the cut and $V \backslash S$ on the $t$-side. Given such a graph, $f$ can be minimized by finding the minimum $s$-$t$ cut set $U^* \subseteq V \cup \mathcal{A}$ in $G$ and taking $S^* = V \cap U^*$. When (2) holds, we say that $G$ *models* $f$.

**Graph reduction strategy for Card-DSFM.** In general, the function $f(S) = \sum_{e \in E} f_e(S \cap e)$ is *not* a concave cardinality function, even if individual components $f_e$ in the sum are. However, in order to show $f$ is graph reducible, it suffices to find a small reduced graph $G_e$ for each $e \in E$ that models $f_e$ in the sense of (2), and combine these together into a larger graph. A number of related approaches for this task have been developed [39, 22, 41]. We review our own previous reduction strategy [41], which we build on in this paper. Modeling $f_e$ can be accomplished by combining a set of *cardinality-based (CB) gadgets* [41]. For a set $e \subseteq V$ of size $k = |e|$, a CB-gadget is a small graph parameterized by positive weights $(a, b)$. Each $v \in e$ defines a node, and the gadget also involves a single auxiliary node $v_e$. For each $v \in e$, there is a directed edge $(v, v_e)$ with weight $a \cdot (k - b)$, and a directed edge $(v_e, v)$ with weight $a \cdot b$. The resulting graph gadget models the function

$$f_{a,b}(S) = a \cdot \min\{|S| \cdot (k - b), (k - |S|) \cdot b\}. \tag{3}$$

To see why, consider where we must place the auxiliary node $v_e$ when solving a minimum $s$-$t$ cut problem involving this small graph gadget. If we place $i = |S|$ nodes on the $s$-side, then placing $v_e$ on the $s$-side has a cut penalty of $ab(k - i)$, whereas placing $v_e$ on the $t$-side gives a penalty of $ai(k - b)$. To minimize the cut as in (2) for a fixed $S \subseteq e$, we choose the smaller of the two cut penalties. We previously showed that any cardinality-based submodular function $f_e$ on a ground set $e$ can be modeled by combining $|e| - 1$ CB-gadgets [41]. Analogously, Stobbe and Krause [39] showed that a concave cardinality function on a ground set $e$ can be decomposed into a sum of $|e| - 1$ *threshold potentials* [39] and can be represented by graph cuts, though they provided no explicit

reduction strategy. Priori to this, Kohli et al. [22] highlighted a similar type of gadget for modeling a class of *robust potential* functions for image segmentation, noting that multiple gadgets could be combined to model arbitrary concave functions. Using any of these techniques, the function $f_e$ can be modeled in the sense of (2) using a graph $G$ with $O(|e|)$ nodes and $O(|e|^2)$ edges.

# 3  Sparse Reductions via Piecewise Linear Approximation

We now turn to a new and refined problem of finding *sparse* and *approximate* reduction strategies for Card-DSFM, which we prove can be cast as approximating a concave function with a piecewise linear curve. All proofs are given in the supplement, where we also derive a similar graph reduction scheme that is slightly more efficient for modeling symmetric functions, i.e., $f_e(A) = f_e(e \backslash A)$.

## 3.1  The sparse approximate reduction problem

We first introduce a special parameterized function with broad modeling power.

**Definition 1** *A $k$-node combined gadget function (CGF) of order $J$ is a function of the form:*

$$\ell(x) = z_0 \cdot (k - x) + z_k \cdot x + \sum_{j=1}^{J} a_j \min\{x \cdot (k - b_j), (k - x) \cdot b_j\}, \qquad (4)$$

*parameterized by scalars $z_0 \geq 0$, $z_k \geq 0$, and vectors $\boldsymbol{a} = (a_j) \in \mathbb{R}_{>0}^J$ and $\boldsymbol{b} = (b_j) \in \mathbb{R}_{>0}^J$, where $b_j < b_{j+1}$ for $j \in \{1, 2, \ldots, J-1\}$ and $b_J < k$.*

If we define a set function on a ground set $e$ of size $k$ by $f_e(S) = \ell(|S|)$, then $f_e$ is exactly the function that is modeled by combining $J$ CB-gadgets with parameters $(a_j, b_j)_{j=1}^J$, and additionally placing a directed edge from a source node $s$ to each $v \in e$ of weight $z_0$, and an edge from $v \in e$ to a sink node $t$ with weight $z_k$. Previous reduction techniques amount to proving that any concave function $g$ on an interval $[0, k]$ (representing a submodular function $f_e(S) = g(|S|)$) matches some $k$-node CGF of order $k - 1$ at integer inputs [41, 39]. We focus on a new sparse reduction problem.

**Definition 2** *Let $g \colon [0, k] \to \mathbb{R}^+$ be a nonnegative concave function and fix $\varepsilon \geq 0$. The sparsest approximate reduction (SpAR) problem seeks a $k$-node CGF $\ell$ with minimum order $J$ so that*

$$g(i) \leq \ell(i) \leq (1 + \varepsilon)g(i) \text{ for all } i \in \{0, 1, 2, \ldots k\}. \qquad (5)$$

This is equivalent to finding the minimum number of CB-gadgets needed to approximately model a concave cardinality function. Thus, solving this problem provides the sparsest reduction in terms of a standard reduction strategy. Without loss of generality, Definition 2 is restricted to considering functions that upper bound $g$. Any other approximating function could be scaled to produce an upper bound leading to the same guarantees for Card-DSFM. Importantly, we care about approximating the concave function $g$ only at integer inputs, since we are ultimately concerned with modeling the set function $f_e(S) = g(|S|)$. Satisfying $g(x) \leq \ell(x) \leq (1 + \varepsilon)g(x)$ for all $x \in [0, k]$ is a much stronger requirement and may require more CB-gadgets than is actually necessary to approximate $f_e$.

**Connection to piecewise linear approximation**  Our first result establishes a precise one-to-one correspondence between a certain class of piecewise linear functions and combined gadget functions.

**Lemma 1** *The $k$-node CGF in (4) is nonnegative, piecewise linear, concave, and has exactly $J + 1$ linear pieces. Conversely, let $\ell' \colon [0, k] \to \mathbb{R}^+$ be concave and piecewise linear with $J + 1$ linear pieces, and let $m_i$ be the slope of the $i$th linear piece and $B_i$ be the $i$th breakpoint. Then $\ell'$ is uniquely characterized as the $k$-node CGF parameterized by $a_i = \frac{1}{k}(m_i - m_{i+1})$ and $b_i = B_i$ for $i \in \{1, 2, \ldots J\}$, $z_0 = \ell'(0)/k$, and $z_k = \ell'(k)/k$.*

This result tells us that solving SpAR (Def 2) for $g$ is equivalent to finding a concave piecewise linear function with the smallest number of linear pieces approximating $g$ at integer points. Although some techniques for approximating a concave function with a piecewise linear curve already exist [34] and could be used as heuristics, these are ultimately unable to find optimal solutions for SpAR. First of all, these methods focus on approximating concave functions over continuous intervals rather than just at integer inputs. More importantly, they do not provide instance optimal approximations, but only give upper bounds on the number of linear pieces needed. We therefore turn to new methods that will allow us to exactly solve our sparse approximation problem.

---

**Algorithm 1** GREEDYPLCOVER$(g, \varepsilon)$

---

**Input**: $\varepsilon \geq 0$, concave function $g$
**Output**: piecewise linear $\ell$ with fewest linear pieces such that $g(i) \leq \ell(i) \leq (1 + \varepsilon)g(i)$
$\mathcal{L} \leftarrow \emptyset, u \leftarrow 0$   //u = smallest integer not covered by approximating line
**while** $u \leq k$ **do**
   $u', L \leftarrow$ NEXTLINE$(g, u, \varepsilon)$   //line covering widest range [u, u'-1]
   $\mathcal{L} \leftarrow \mathcal{L} \cup \{L\}, u \leftarrow u'$
**end while**
Return $\ell(x) = \min_{L \in \mathcal{L}} L(x)$

---

## 3.2 Optimal sparse approximate reduction

Our goal is to find a piecewise linear function $\ell$ with a minimum number of linear pieces satisfying condition (5). This is equivalent to finding a minimum-sized collection $\mathcal{L}$ of linear functions that "cover" points $\{j, g(j)\}$ for integers $j$, in the sense that (i) each linear function $L \in \mathcal{L}$ satisfies $g(j) \leq L(j)$ for all $j \in \{0, 1, \ldots, k\}$, and (ii) for every $j \in \{0, 1, \ldots, k\}$ there exists some $L \in \mathcal{L}$ satisfying $L(j) \leq (1 + \varepsilon)g(j)$. Then, $\ell(x) = \min_{L \in \mathcal{L}} L(x)$ is the desired piecewise linear solution.

We develop a simple method (Alg. 1) to grow a collection $\mathcal{L}$ satisfying these conditions. Each iteration considers the integer $u$ that is not currently covered by a $(1 + \varepsilon)$-approximating line, and uses a subroutine NEXTLINE to find a line $L$ that covers $u$ and also satisfies $g(t) \leq L(t) \leq (1 + \varepsilon)g(t)$ for the largest integer $t$. This is done by taking the line through the point $\{u, (1 + \varepsilon)g(u)\}$ that has the minimum slope while still upper bounding every point $\{i, g(i)\}$. To provide intuition for this strategy, note that $\mathcal{L}$ must contain *some* line that provides a $(1+\varepsilon)$-approximation at $\{0, g(0)\}$. It can only help us to choose a line that provides this guarantee while also providing a $(1 + \varepsilon)$-approximation for the widest possible range $\{0, 1, \ldots, t\}$. The same logic applies to finding linear pieces to approximate the function at remaining integer inputs. The supplementary material contains pseudocode for NEXTLINE and a proof of the following result.

**Theorem 2** *Algorithm 1 solves the sparsest approximate reduction problem in $O(k)$ operations.*

## 3.3 Bounds on optimal reduction size

For a concave function $g$, an existing method of Magnanti and Stratila [33, 34] can find a piecewise linear function that approximates $g$ over a continuous interval $[a, b]$ with $O(\frac{1}{\varepsilon} \log \frac{b}{a})$ linear pieces. Although this method does not optimally solving SpAR, it can be used to show the following worst-case upper bound on the number of linear pieces found by our instance-optimal method.

**Theorem 3** *Let $g \colon [0, k] \to \mathbb{R}^+$ be concave and let $\varepsilon \geq 0$. Algorithm 1 will return a piecewise linear function $\ell$ with at most $O(\min\{k, \frac{1}{\varepsilon} \log k\})$ linear pieces satisfying (5).*

Previous lower bounds on the number of linear pieces needed to approximate a concave function do not apply to our problem, as these are focused on approximating functions over continuous intervals [34]. Since $k$ points on a concave function can always be covered using $k$ linear pieces, proving meaningful lower bounds for our problem is more challenging. Nevertheless, using an existing lower-bound for approximating the square root function over a continuous interval as a black-box [34], we prove that the upper bound in Theorem 3 is nearly asymptotically tight.

**Theorem 4** *Let $g(x) = \sqrt{x}$ and $\varepsilon \geq k^{-\delta}$ for a constant $\delta \in (0, 2)$. Every piecewise linear $\ell$ satisfying $g(i) \leq \ell(i) \leq (1 + \varepsilon)g(i)$ for $i \in \{0, 1, \ldots k\}$ contains $\Omega(\log_{\gamma(\varepsilon)} k)$ linear pieces where $\gamma(\varepsilon) = (1 + 2\varepsilon(2 + \varepsilon) + 2(1 + \varepsilon)\sqrt{\varepsilon(2 + \varepsilon)})^2$. This behaves as $\Omega(\varepsilon^{-1/2} \log k)$ as $\varepsilon \to 0$.*

In many cases the number of linear pieces returned by Algorithm 1 will be far smaller than the bounds above and the number of pieces returned by previous approaches [34]. One of our central contributions is the following guarantee for a concave function that arises extensively in applications. Its proof is somewhat involved, and relies on bounding the number of iterations it takes to provide an approximation for subintervals of the form $[k\varepsilon^{1/2^{j-1}}, k\varepsilon^{1/2^j}]$ for $j = 1$ to $j = \lceil \log_2 \log_2 \varepsilon^{-1} \rceil$.

**Algorithm 2** SPARSECARD($f, \varepsilon$)

---

**Input**: $\varepsilon \geq 0$, function $f(S) = \sum_{e \in E} f_e(S \cap e) = \sum_{e \in E} g_e(|S \cap e|)$ on ground set $V$
**Output**: Set $S' \subseteq V$ satisfying $f(S') \leq (1 + \varepsilon) \min_{S \subseteq V} f(S)$.
$\mathcal{A} \leftarrow \emptyset, \mathcal{E} \leftarrow \emptyset$   //initialize auxiliary node and edge set for reduced graph
**for** $e \in E$ **do**
    $\ell_e \leftarrow$ GREEDYPLCOVER$(g_e, \varepsilon)$            //1: Solve SpAR (Algorithm 1)
    $G_e = (e \cup \mathcal{A}_e, \mathcal{E}_e) \leftarrow$ CGFTOGADGET$(\ell_e)$ //2: Build combined gadget (Lemma 1)
    $\mathcal{A} \leftarrow \mathcal{A} \cup \mathcal{A}_e, \mathcal{E} \leftarrow \mathcal{E} \cup \mathcal{E}_e$         //3: Add gadget to graph
**end for**
$G = (V \cup \mathcal{A} \cup \{s, t\}, \mathcal{E})$           //Build graph G modeling f
$T = $ MINSTCUT$(G)$                 //Find minimum s-t cut
Return $S' = T \cap V$             //Ignore auxiliary nodes

---

**Theorem 5** *Let $g(x) = x \cdot (k - x)$ for a positive integer $k$. For $\varepsilon > 0$, the approximating function $\ell$ returned by Algorithm 1 will have $O(\varepsilon^{-1/2} \log \log \varepsilon^{-1})$ linear pieces.*

This result is significant in a number of ways. First of all, and somewhat surprisingly, the number of linear pieces needed to approximate $g$ is independent of $k$. Our instance optimal algorithm therefore leads to a significant improvement over the $O(\varepsilon^{-1} \log k)$ upper bound in Theorem 3 and the piecewise linear approximation that could be obtained using existing techniques [34]. More importantly, this theorem has significant implications for approximately modeling the concave cardinality function $f_e(S) = c_e |S||e \backslash S|$ (where $c_e$ can be any positive constant) using graph cuts. Although previous exact graph reduction techniques [22, 39, 42] require $O(|e|)$ nodes and $O(|e|^2)$ edges to model this function, we can approximate it with only $O(|e|\varepsilon^{-1/2} \log \log \varepsilon^{-1})$ edges. This savings is particularly significant given how extensively this function appears in practice. This function is a commonly used *region potential* in image segmentation, and shows up frequently in work on DSFM [9, 19, 39, 29]. It also appears often in hypergraph clustering applications, where it results from a *clique expansion* of a hypergraph [2, 1, 7, 16, 47], which replaces each hyperedge $e$ with a clique on nodes in $e$. The cut function of the resulting graph is a decomposable submodular function whose components have the form $f_e(S) = c_e |S||e \backslash S|$. This arises similarly as the cut function of a graph obtained from co-occurrence relationships, or a graph obtained from a projection of a bipartite graph [8]. Our results provide a useful new type of sparsifier for all of these types of graphs.

Theorem 5 also has implications for sparsifying a complete graph, a problem of interest in the theoretical computer science literature. Existing sparsifiers model the cut properties of a complete graph on $n$ nodes with $O(n\varepsilon^{-2})$ edges [6]. This is tight for spectral sparsifiers [6], as well as for degree-regular cut sparsifiers with uniform edge weights [3]. Our result implies that if we are willing to include a small number of additional nodes and use directed edges, the cut properties of the complete graph can be modeled using only $O(n)$ nodes and $O(n\varepsilon^{-1/2} \log \log \varepsilon^{-1})$ edges.

## 4 Theoretical Runtime Analysis and Comparison

The ability to approximately model a single concave cardinality function makes it possible to quickly obtain an arbitrarily good approximate solution to an instance of Card-DSFM by reducing it to a minimum $s$-$t$ cut problem on a sparse graph. Define a function $f$ on a ground set $V$ by

$$f(S) = \sum_{e \in E} f_e(S \cap e) = \sum_{e \in E} g_e(|S \cap e|), \tag{6}$$

where each $g_e$ is concave. We assume each $f_e$ is nonnegative; if not we can adjust the objective by a constant without affecting optimal solutions. Algorithm 2 gives pseudocode for our new method SPARSECARD for minimizing (6). The method finds a sparse approximate graph reduction for each concave function $g_e$ using Algorithm 1, combines these into a larger graph whose cut properties approximate $f$, and then applies a minimum $s$-$t$ cut solver to that graph. Finding sparse reductions is fast, so the asymptotic runtime guarantee is just the time it takes to solve the cut problem, which can be accomplished using any algorithm for solving the dual maximum $s$-$t$ flow problem [15, 14, 26, 40].

**Theorem 6** *Let $n = |V|$ and $R = |E|$. When $\varepsilon = 0$, the graph constructed by SPARSECARD will have $O(\sum_{e \in E} |e|)$ nodes and $O(\sum_{e \in E} |e|^2)$ edges. When $\varepsilon > 0$, the graph will have $O(n +$*

$\sum_{e \in E} \varepsilon^{-1} \log |e|)$ *nodes and* $O(\varepsilon^{-1} \sum_{e \in E} |e| \log |e|)$ *edges. In either case, and the method will return a set $T$ satisfying $f(T) \leq (1 + \varepsilon) \min_{S \subseteq V} f(S)$.*

The graph returned by SPARSECARD can also be asymptotically sparser for specialized concave functions, such as the popular clique expansion function in Theorem 5.

In the remainder of the section, we provide a careful runtime comparison between SPARSECARD and competing runtimes for Card-DSFM. We reiterate that most previous techniques are designed to solve more general DSFM problems. However, these typically achieve improved runtime guarantees in the case of cardinality-based components. Our goal is to understand the theoretical runtime improvements that can be achieved for Card-DSFM, especially in cases where approximate solutions suffice. We focus on each runtime's dependence on $n = |V|$, $R = |E|$, and support sizes $|e|$, and use $\tilde{O}$ notation to hide logarithmic factors of $n$, $R$, and $1/\varepsilon$. To easily compare weakly polynomial runtimes, we assume that each $f_e$ has integer outputs, and assume that $\log(\max_S f(S))$ is small enough that it can also be absorbed by $\tilde{O}$ notation. Among algorithms for DSFM, SPARSECARD is unique in its ability to quickly find solutions with a priori multiplicative approximation guarantees. Previous approaches for DSFM focus on either obtaining exact solutions, or solving a problem to within an *additive* approximation error $\epsilon > 0$ [4, 9, 29, 19]. In the latter case, setting $\epsilon$ small enough will guarantee an optimal solution in the case of integer output functions. We stress that this type of approximation is very different in nature and not directly comparable to the multiplicative approximations achieved by SPARSECARD. In our comparisons, we assume that previous methods are run until optimality, since the runtime of competing methods only improves by logarithmic factors if we assume the method is run until an additive approximation error $\epsilon > 0$ is achieved.

**Competing runtime guarantees.** Table 1 lists runtimes for existing methods for DSFM. We also give the asymptotic runtime for SPARSECARD when applying the recent maximum flow algorithm of van den Brand et al. [40]. While this leads to the best theoretical guarantees for our method, asymptotic runtime improvements over competing methods can also be shown using alternative fast algorithms for maximum flow [14, 26, 15]. For the submodular flow algorithm of Kolmogorov [24], we have reported the runtime guarantee provided specifically for Card-DSFM. While other approaches have frequently been applied to Card-DSFM [9, 39, 29, 19], runtimes guarantees for this case have not been presented explicitly and are more challenging to exactly pinpoint. Runtimes for most algorithms depend on certain oracles for solving smaller minimization problems at functions $f_e$ in an inner loop. For $e \in E$, let $\mathcal{O}_e$ be a *quadratic minimization oracle*, which for an arbitrary vector $w$ solves $\min_{y \in B(f_e)} \|y + w\|$ where $B(f_e)$ is the base polytope of the submodular function $f_e$ (see [9, 5, 19] for details). Let $\theta_e$ be the time it take to evaluate the oracle at $e \in E$, and define $\theta_{max} = \max_{e \in E} \theta_e$ and $\theta_{avg} = \frac{1}{R} \sum_{e \in E} \theta_e$. Although these oracles admit faster implementations in the case of concave cardinality functions, it is not immediately clear from previous work what is the best possible runtime. When $w = 0$, solving $\min_{y \in B(f_e)} \|y + w\|$ takes $O(|e| \log |e|)$ time [19], so this serves as a best case runtime we can expect for the more general oracle $\mathcal{O}_e$ based on previous results. We note also that in the case of the region potential function $f_e(A) = |A||e \backslash A|$, Ene et al. [9] highlight that an $O(|e| \log |e| + |e|\tau_e)$ algorithm can be used, where $\tau_e$ denotes the time it take to evaluate $f_e(S \cap e)$ for any $S \subseteq e$. In our runtime comparisons will use the bound $\theta_e = \Omega(|e|)$, as it is reasonable to expect that any meaningful submodular function we consider should take a least linear time to minimize. We remark finally that for discrete DSFM algorithms, a slightly weaker oracle than the quadratic minimization oracle can be used [4, 9]. However, this does not lead to improved runtime guarantees, as these are subject to the same lower bounds in the cardinality-based case.

**Fast approximate solutions** ($\varepsilon > 0$)**.** Barring the regime where support sizes $|e|$ are all very small, the accelerated coordinate descent method (ACDM) of Ene et al. [10] provides the fastest previous runtime guarantee. For a simple parameterized runtime analysis, consider a DSFM problem where the average support size is $(1/R) \sum_e |e| = \Theta(n^\alpha)$ for $\alpha \in [0, 1]$, and $R = \Theta(n^\beta)$, where $\beta \geq 1 - \alpha$ must hold if we assume each $v \in V$ is in the support for at least one function $f_e$. An exact runtime comparison between SPARSECARD and ACDM depends on the best runtime for the oracle $\mathcal{O}_e$ for concave cardinality functions. If an $O(|e| \log |e|)$ oracle is possible, the overall runtime guarantee for ACDM would be $\Omega(n^{1+\alpha+\beta})$. Meanwhile, for a small constant $\varepsilon > 0$, SPARSECARD provides a $(1 + \varepsilon)$-approximate solution in time $\tilde{O}(n^{\alpha+\beta} + \max\{n^{3/2}, n^{3\beta/2}\})$, which will faster by at least a factor $\tilde{O}(\sqrt{n})$ whenever $\beta \leq 1$. When $\beta > 1$, finding an approximation with SPARSECARD is guaranteed to be faster whenever $R = o(n^{2+2\alpha})$. If the best case oracle $\mathcal{O}_e$ for concave cardinality functions is $\omega(|e| \log |e|)$, the runtime improvement of our method is even more significant.

Table 1: Runtimes for Card-DFSM for various methods, where $R = |E|$, $n = |V|$, $\mu = \sum_e |e|$, and $\mu_2 = \sum_e |e|^2$. Oracle runtimes satisfy $\theta_{max} = \Omega(\max |e|)$, and $\theta_{avg} = \Omega(\frac{1}{R} \sum_{e \in E} |e|)$. $T_{mf}(N, M)$ is the time to solve a max-flow problem with $N$ nodes and $M$ edges.

| Method | Discrete/Cont | Runtime |
|---|---|---|
| Kolmogorov SF [24] | Discrete | $\tilde{O}(\mu^2)$ |
| IBFS [11, 9] | Discrete | $\tilde{O}(n^2 \theta_{max} + n \sum_e |e|^4)$ |
| AP [35, 9, 29] | Continuous | $\tilde{O}(nR\theta_{avg}\mu)$ |
| RCDM [10, 9] | Continuous | $\tilde{O}(n^2 R\theta_{avg})$ |
| ACDM [10, 9] | Continuous | $\tilde{O}(nR\theta_{avg})$ |
| Axiotis et al. [4] | Discrete | $\tilde{O}(\max_e |e|^2 \cdot (\sum_e |e|^2 \theta_e + T_{mf}(n, n + \mu_2)))$ |
| SPARSECARD $\varepsilon = 0$ | Discrete | $\tilde{O}(T_{mf}(\mu, \mu_2)) = \tilde{O}(\mu_2 + \mu^{3/2})$ |
| SPARSECARD $\varepsilon > 0$ | Discrete | $\tilde{O}(T_{mf}(n + \frac{R}{\varepsilon}, \frac{1}{\varepsilon}\mu)) = \tilde{O}(\frac{\mu}{\varepsilon} + (n + \frac{R}{\varepsilon})^{3/2})$ |

**Guarantees for exact solutions ($\varepsilon = 0$).** As an added bonus, running SPARSECARD with $\varepsilon = 0$ leads to the fastest runtime for finding *exact* solutions in many regimes. In this case, we can guarantee SPARSECARD will be faster than ACDM when the average support size is $\Theta(n^\alpha)$ and $R = o(n^{2-\alpha})$. SPARSECARD can also find exact solutions faster than other discrete optimization methods [24, 4, 11] in wide parameter regimes. Unlike SPARSECARD, these methods are designed for problems where all support sizes are small, but become impractical if even a single function has a large support size.

The runtime guarantee for SPARSECARD when $\varepsilon = 0$ can be matched asymptotically by combining existing exact reduction techniques [22, 39, 41] with fast maximum flow algorithms. However, our method has the practical advantage of finding the *sparsest* exact reduction in terms of CB-gadgets. This results in a reduced graph with roughly half the number of edges as our previous exact reduction technique [41]. Analogously, while Stobbe and Krause [39]) showed that a concave cardinality function can be decomposed as a sum of modular functions plus a combination of $|e| - 1$ threshold potentials, our approximation technique will find a linear combination with $\lfloor |e|/2 \rfloor$ threshold potentials. This amounts to the observation that any $k + 1$ points $\{i, g(i)\}$ can be joined by $\lfloor k/2 \rfloor + 1$ lines instead of using $k$. Overall though, the most significant advantage of SPARSECARD over existing reduction methods is its ability to find fast approximate solutions with sparse approximate reductions.

## 5 Experiments

In addition to its strong theoretical guarantees, SPARSECARD is very practical and leads to substantial improvements in benchmark image segmentation problems and hypergraph clustering tasks. We focus on DSFM problems that simultaneously include component functions of large and small support, which are common in computer vision and hypergraph clustering applications [38, 9, 41, 32, 37]. We ran experiments on a laptop with a 2.2 GHz Intel Core i7 processor and 8GB of RAM. We consider public datasets previously made available for academic research, and use existing open source software for competing methods.[2] Code for our algorithms and experimental results is available at https://github.com/nveldt/SparseCardDSFM.

**Benchmark Image Segmentation Tasks.** SPARSECARD provides faster approximate solutions for standard image segmentation tasks previously used as benchmarks for DSFM [19, 29, 9]. We consider the *smallplant* and *octopus* segmentation tasks from Jegelka et al. [20, 19]. These amount to minimizing a decomposable submodular function on a ground set of size $|V| = 427 \cdot 640 = 273280$, where each $v \in V$ is a pixel from a $427 \times 640$ pixel image and there are three types of component functions. The first type are unary potentials for each pixel/node, i.e., functions of support size 1 representing each node's bias to be in the output set. The second type are pairwise potentials from a 4-neighbor grid graph; pixels $i$ and $j$ share an edge if they are directly adjacent vertically or horizontally. The third type are region potentials of the form $f_e(A) = |A||e \backslash A|$ for $A \subseteq e$,

---

[2]Image datasets: http://people.csail.mit.edu/stefje/code.html. Hypergraph clustering datasets: www.cs.cornell.edu/~arb/data/. DSFM algorithms: from github.com/lipan00123/DSFM-with-incidence-relations (MIT license); Hypergraph clustering algorithms: github.com/nveldt/HypergraphFlowClustering (MIT license).

Table 2: Results from SPARSECARD for different $\varepsilon > 0$ on the *smallplant* instance with 500 superpixels. Sparsity is the fraction of edges in the approximate graph reduction compared with the exact reduction. Finding the exact solution on the dense exact reduced graph took $\approx 20$ minutes.

| $\varepsilon$ | 1.0 | 0.2336 | 0.0546 | 0.0127 | 0.003 | 0.0007 | 0.0002 |
|---|---|---|---|---|---|---|---|
| Approx.$-1$ | $4 \cdot 10^{-3}$ | $2 \cdot 10^{-3}$ | $6 \cdot 10^{-4}$ | $6 \cdot 10^{-5}$ | $3 \cdot 10^{-5}$ | $7 \cdot 10^{-6}$ | $7 \cdot 10^{-7}$ |
| Sparsity | 0.013 | 0.017 | 0.02 | 0.035 | 0.06 | 0.108 | 0.196 |
| Runtime | 4.1 | 5.6 | 6.7 | 11.5 | 24.3 | 41.4 | 74.3 |

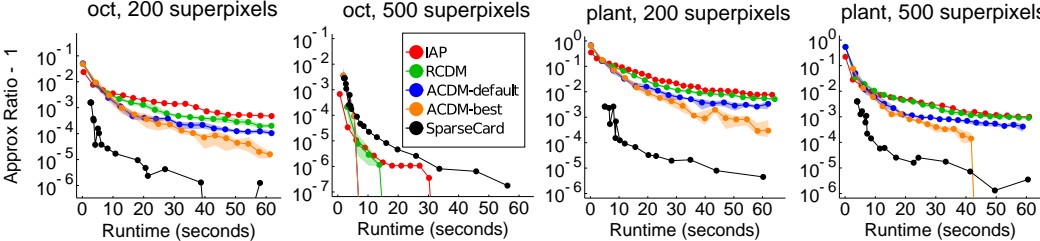

Figure 1: Approximation factor minus 1 vs. runtime for solutions returned by SPARSECARD and competing methods on four image segmentation tasks. We display the average of 5 runs for competing methods, with lighter colored region showing upper and lower bounds from these runs. SPARSECARD is deterministic and was run once for each $\varepsilon$ on a decreasing logarithmic scale. Our method maintains an advantage even against post-hoc best case parameters for competing approaches: ACDM-best is the best result obtained by running ACDM for a range of empirical parameters $c$ for each dataset and reporting the best result. The default is $c = 10$ (blue curve). Best post-hoc results for the plots from left to right were $c = 25, 10, 50, 25$. It is unclear how to determine the best $c$ in advance.

where $e$ represents a superpixel region. The problem can be solved via maximum flow even without sophisticated reduction techniques for cardinality functions, as a regional potential function on $e$ can be modeled by placing a clique of edges on $e$. We compute an optimal solution using this reduction. Compared with the exact reduction method, running SPARSECARD with $\varepsilon > 0$ leads to much sparser graphs, much faster runtimes, and a posteriori approximation factors that are significantly better than $(1 + \varepsilon)$. In Table 2 we list the sparsity, runtime, and a posteriori guarantee obtained for a range of $\varepsilon$ values on the *smallplant* dataset using the superpixel segmentation with 500 regions.

We also compare against recent C++ implementations of ACDM, RCDM, and Incidence Relation AP (an improved version of the standard AP method [35]) provided by Li and Milenkovic [29]. These use the divide-and-conquer method of Jegelka et al. [19], implemented specifically for concave cardinality functions, to solve the quadratic minimization oracle $\mathcal{O}_e$ for region potential functions. Although these continuous optimization methods come with no a priori approximation guarantees, we can compare them against SPARSECARD by computing a posteriori approximations obtained using intermediate solutions returned after every few hundred iterations. Figure 1 displays approximation ratio versus runtime for four DSFM instances (two datasets $\times$ two superpixel segmentations). SPARSECARD was run for a range of $\varepsilon$ values on a decreasing logarithmic scale from 1 to $10^{-4}$, and obtains significantly better results on all but the *octopus* with 500 superpixels instance. This is the easiest instance; all methods obtain a solution within a factor 1.001 of optimality within a few seconds. ACDM depends on a hyperparameter $c$ controlling the number of iterations in an outer loop. Even when we choose the best post-hoc $c$ value for each dataset, SPARSECARD maintains its overall advantage. Note that we focus on comparisons with continuous optimization methods rather than other discrete optimization methods, as the former are better equipped for our goal of finding approximate solutions to DSFM problems involving functions of large support. To our knowledge, no implementations for the methods of Kolmogorov [24] or Axiotis et al. [4] exist. Meanwhile, IBFS [11] is designed for finding exact solutions when all support sizes are small. Recent empirical results [9] confirm that this method is not designed to handle the large region potential functions we consider here.

**Hypergraph local clustering.** Graph reduction techniques have been frequently and successfully used as subroutines for hypergraph local clustering and semi-supervised learning methods [32, 42, 28, 45]. Replacing exact reductions with our approximate reductions can lead to significant runtime improvements without sacrificing on accuracy, and opens the door to running local clustering algorithms on problems where exact graph reduction would be infeasible. We illustrate this by

Table 3: Average F1 score and standard deviation for detecting 45 local clusters in a stackoverflow question hypergraph using HyperLocal [41] + SPARSECARD with four hyperedge cut penalty functions. The $\delta$-linear penalty had the fastest runtime (26 seconds on average) as it has a sparse optimal ($\varepsilon = 0$) reduction. For $\Delta$Time, we compute the ratio between the runtime of $\delta$-linear and the runtime of each method on all 45 clusters, then report the mean and standard deviation of these ratios. The *# Best* row indicates the number of times a method obtains the highest F1 score out of the 45 clusters.

| | $\delta$-linear | clique | | $x^{0.9}$ | | sqrt | |
|---|---|---|---|---|---|---|---|
| | $\varepsilon = 0$ | $\varepsilon = 1$ | $\varepsilon = 0.1$ | $\varepsilon = 1$ | $\varepsilon = 0.1$ | $\varepsilon = 1$ | $\varepsilon = 0.1$ |
| F1 | $0.53 \pm 0.22$ | $0.56 \pm 0.19$ | $0.56 \pm 0.19$ | $0.54 \pm 0.20$ | $0.54 \pm 0.21$ | $0.42 \pm 0.18$ | $0.42 \pm 0.19$ |
| $\Delta$Time | 1 | $1.32 \pm 0.33$ | $1.81 \pm 0.43$ | $1.17 \pm 0.25$ | $2.04 \pm 0.42$ | $2.02 \pm 0.99$ | $3.19 \pm 1.4$ |
| # Best | 7 | 10 | 16 | 8 | 3 | 0 | 1 |

using SPARSECARD as a subroutine for an existing method called HYPERLOCAL [42]. This algorithm finds local clusters in a hypergraph by repeatedly solving Card-DSFM problems corresponding to hypergraph minimum $s$-$t$ cuts. For these DSFM problems, $e \in E$ is a hyperedge and $f_e(A)$ is the penalty for cutting a hyperedge so that nodes in $A \subseteq e$ are on one side of the cut. HYPERLOCAL was originally designed to handle only the $\delta$-*linear* penalty $f_e(A) = \min\{|A|, |e \setminus A|, \delta\}$, for parameter $\delta \geq 1$, which can already be modeled sparsely with a single CB-gadget. SPARSECARD makes it possible to sparsely model any concave cardinality penalty. We specifically use approximate reductions for the weighted clique penalty $f_e(A) = (|e| - 1)^{-1}|A||e \setminus A|$, the square root penalty $f_e(A) = \sqrt{\min\{|A|, |e \setminus A|\}}$, and the sublinear power function penalty $f_e(A) = (\min\{|A|, |e \setminus A|\})^{0.9}$, all of which require $O(|e|^2)$ edges to model exactly using previous reduction techniques. Weighted clique penalties in particular have been used extensively in hypergraph clustering [1, 45, 25, 47], including by methods specifically designed for local clustering and semi-supervised learning [27, 45, 46].

We consider a hypergraph clustering problem where nodes are 15.2M questions on `stackoverflow.com` and each of the 1.1M hyperedges defines a set of questions answered by the same user. The mean hyperedge size is 23.7, the maximum size is over 60k, and there are 2165 hyperedges with at least 1000 nodes. Questions with the same topic tag (e.g., "common-lisp") constitute small labeled clusters in the dataset. We previously showed that HYPERLOCAL can detect clusters quickly with the $\delta$-linear penalty by solving localized $s$-$t$ cut problems near a seed set. Applying exact graph reductions for other concave cut penalties is infeasible, due to the extremely large hyperedge sizes, and we found previously that using a clique expansion after simply removing large hyperedges performs poorly [42]. Using SPARSECARD as a subroutine opens up new possibilities.

Following an existing approach [42], we seek to detect 45 labeled clusters using a random seed set of 5% of each cluster. Table 1 reports average F1 scores and relative runtimes for four hyperedge cut penalties. Given the natural variation in cluster structure and size, standard deviations should not be viewed as error bars for each approach per se, but these provide a rough indication for how the performance of each method varies across clusters. Detailed results for each cluster are included in the supplement. Importantly, cut penalties that previously could not be used on this dataset (*clique*, $x^{0.9}$) obtain the best results for most clusters. The square root penalty does not perform particularly well on this dataset, but it is instructive to consider its runtime. Theorem 4 shows that asymptotically this function has a worst-case behavior in terms of the number of CB-gadgets needed to approximate it. We nevertheless obtain reasonably fast results for this penalty function, indicating that our techniques can provide effective sparse reductions for any concave cardinality function of interest. We also ran experiments with $\varepsilon = 0.01$, which led to noticeable increases in runtime but only very slight changes in F1 scores. This indicates why exact reductions are not possible in general, while also showing that our sparse approximate reductions serve as fast and very good proxies for exact reductions.

## 6  Conclusion and Discussion

We have introduced a sparse graph reduction technique leading to the first approximation algorithms for cardinality-based DSFM. Our method provides an optimal reduction strategy in terms of previously considered graph gadgets, comes with improved theoretical runtime guarantees over competing methods, and leads to significant improvements in benchmark image segmentation and hypergraph clustering experiments. An interesting direction for future research is to explore lower bounds or

improved techniques for other possible graph reduction strategies. For the very special case of clique splitting functions, a practical open question to explore is whether uniformly sampling edges in a clique works well in practice. On a more theoretical note, another open question to explore is whether the maximum flow techniques of Axiotis et al. [4] could be combined with our sparsification methods to achieve even better theoretical runtimes for cardinality-based DSFM.

Regarding potential limitations of our work, our method applies only to the cardinality-based variant of the problem, whereas most existing methods solve a more general problem. Nevertheless, Card-DSFM is one of the most widely applied variants in practice, which highlights the utility of developing better theory and algorithms for this special case. Another limitation is that our method is not as easy to parallelize as continuous optimization methods. An open question is whether better specialized (parallel or serial) runtimes can be obtained for continuous methods for Card-DSFM. Finally, while our research focuses on faster algorithms for a fundamental optimization task, there are ways in which tools for image segmentation and clustering (which are downstream applications of our work) can result in negative societal impacts depending on their use. For example, image segmentation could be used in illicit targeted video surveillance. Clustering methods could be used to de-anonymized private information in a social network, or to segment a population of voters for micro-targeted political campaigns that potentially lead to increased political polarization. Nevertheless, algorithms for decomposable submodular function minimization, as well as the more specific tasks of image segmentation and hypergraph clustering, remain very general and are also broadly useful for many positive applications.

## Funding Statement

Austin Benson is supported by ARO Award W911NF-19-1-0057, ARO MURI, and NSF CAREER Award IIS-2045555. Jon Kleinberg is supported by a Vannevar Bush Faculty Fellowship, ARO MURI, AFOSR grant FA9550-19-1-0183, and a Simons Investigator grant. We declare no financial competing interests.

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
