# Supplementary Text:
# Approximate Decomposable Submodular Function Minimization for Cardinality-Based Components

**Nate Veldt**
Texas A&M University
nveldt@tamu.edu

**Austin R. Benson**
Cornell University
arb@cs.cornell.edu

**Jon Kleinberg**
Cornell University
kleinberg@cornell.edu

## A   Proofs of Main Theoretical Results

This section includes detailed proofs of all of our main theoretical results for sparse approximate reductions via piecewise linear approximation. For convenience we include definitions and theorem statements from the main text, expanded in some cases to provide additional clarifying details. Recall that our goal is to sparsely (and approximately) model the submodular function

$$f(S) = \sum_{e \in E} f_e(S \cap e) = \sum_{e \in E} g_e(|S \cap e|), \text{ for } S \subseteq V \tag{1}$$

using a graph, which we will do by showing how to sparsely model each component function $f_e$ individually.

**Terminology and notation** For our results it will be convenient to interpret the ground set $V$ as a set of nodes in a hypergraph $\mathcal{H} = (V, E)$, where each $e \in E$ is an individual hyperedge and $f_e$ is the function which determines how to penalize different ways of splitting the hyperedge $e$. The function $f$ is then a notion of a generalized hypergraph cut function [4, 6, 9]. This terminology is particularly convenient when talking about graph reduction strategies, since modeling a function $f_e$ will involve treating each element in $e$ as a node. We apply this terminology here, though note that all of the results we show will apply more generally to approximately modeling a decomposable submodular function using the cut properties of a graph.

Throughout the supplement we use $[d]$ to denote the set $\{1, 2, \cdots, d\}$ for any positive integer $d \in \mathbb{N}$.

### A.1   CB-gadgets and their combinations

We use combinations of cardinality-based (CB) gadgets to model a function $f_e$. Let $k = |e|$. The cut properties of this gadget model the function $f_e(A) = g_e(|A|)$ where

$$g_e(x) = a \cdot \min\{x \cdot (k - b), (k - x) \cdot b\}.$$

The following parameterized concave function corresponds to a combination of multiple CB-gadgets.

**Definition 1** *A $k$-node combined gadget function (CGF) of order $J$ is a function $\ell \colon [0, k] \to \mathbb{R}^+$ of the form.*

$$\ell(x) = z_0 \cdot (k - x) + z_k \cdot x + \sum_{j=1}^{J} a_j \min\{x \cdot (k - b_j), (k - x) \cdot b_j\}. \tag{2}$$

*where $z_0$ and $z_k$ are non-negative parameters, and the $J$-dimensional vectors $\boldsymbol{a} = (a_j)$ and $\boldsymbol{b} = (b_j)$ satisfy the following constraints:*

$$b_j > 0, a_j > 0 \text{ for all } j \in [J] \tag{3}$$
$$b_j < b_{j+1} \text{ for } j \in [J - 1] \tag{4}$$
$$b_J < k. \tag{5}$$

35th Conference on Neural Information Processing Systems (NeurIPS 2021).

The conditions on the vectors **a** and **b** come from natural observations about combining CB-gadgets. Conditions (3) and (5) ensure that we do not consider CB-gadgets where with edge weights that are zero. The ordering in condition (4) is for convenience, and the fact that $b_j$ values are all distinct implies that we cannot collapse two distinct CB-gadgets into a single CB-gadget with new weights.

We now prove the connection between concave functions with few linear pieces and combined gadget functions of small order $J$.

**Lemma 1** *The $k$-node CGF in* (2) *is nonnegative, piecewise linear, concave, and has exactly $J + 1$ linear pieces. Conversely, let $\ell': [0, k] \to \mathbb{R}^+$ be concave and piecewise linear with $J + 1$ linear pieces, and let $m_i$ be the slope of the $i$th linear piece and $B_i$ be the $i$th breakpoint. Then $\ell'$ is uniquely characterized as the $k$-node CGF parameterized by $a_i = \frac{1}{k}(m_i - m_{i+1})$ and $b_i = B_i$ for $i \in \{1, 2, \ldots J\}$, $z_0 = \ell'(0)/k$, and $z_k = \ell'(k)/k$.*

**Proof** We break up the proof into its two directions.

*First direction: the CGF is a special type of piecewise linear function.* Nonnegativity follows quickly from the positivity of $z_0$, $z_k$, and $(a_i, b_i)$ for $i \in [J]$, and $b_J < k$. For other properties, we begin by defining $b_0 = 0$, $b_{J+1} = k$, $a_0 = a_{J+1} = 0$ for notational convenience. This allows us to re-write the function as

$$\ell(x) = z_0 \cdot (k - x) + z_k \cdot x + \sum_{j=1}^{J} a_j \min\{x \cdot (k - b_j), (k - x) \cdot b_j\} \tag{6}$$

$$\ell(x) = z_0 \cdot (k - x) + z_k \cdot x + \sum_{j=0}^{J+1} a_j \min\{x \cdot (k - b_j), (k - x) \cdot b_j\} \tag{7}$$

$$= kz_0 + x(z_k - z_0) + k \cdot \sum_{j=0}^{J+1} a_j \min\{x, b_j\} - x \cdot \sum_{j=1}^{J} a_j b_j \tag{8}$$

$$= kz_0 + x(z_k - z_0) + kx \cdot \sum_{j:x<b_j} a_j + k \cdot \sum_{j:x \geq b_j} a_j b_j - x \cdot \sum_{j=1}^{J} a_j b_j. \tag{9}$$

Now for $t \in \{0\} \cup [J]$ we define

$$\beta = \sum_{j=1}^{J} a_j b_j, \qquad \beta_t = \sum_{j=1}^{t} a_j b_j, \qquad \alpha_t = \sum_{j=t+1}^{J+1} a_j,$$

and observe that $\beta_t$ is strictly increasing with $t$, and $\alpha_t$ is strictly *decreasing* with $t$. For any $t \in \{0\} \cup [J]$, the function is linear over the interval $[b_t, b_{t+1})$, since for $x \in [b_t, b_{t+1})$, we have

$$\ell(x) = kz_0 + x(z_k - z_0) + kx \cdot \sum_{j:x<b_j} a_j + k \cdot \sum_{j:x \geq b_j} a_j b_j - x \sum_{j=1}^{J} a_j b_j$$

$$= kz_0 + x(z_k - z_0) + kx \sum_{j=t+1}^{J+1} a_j + k \cdot \sum_{j=1}^{t} a_j b_j - x \sum_{j=1}^{J} a_j b_j$$

$$= kz_0 + x(z_k - z_0) + kx\alpha_t + k\beta_t - x\beta.$$

Thus, $\ell$ is piecewise linear. Furthermore, the slope of the line over the interval $[b_t, b_{t+1})$ is $(z_k - z_0 - \beta + k\alpha_t)$, which strictly decreases as $t$ increases. The fact that all slopes are distinct means that there are exactly $J + 1$ linear pieces, and the fact that these slopes are decreasing means that the function is concave over the interval $[0, k]$.

*Second direction: $\ell'$ is a CGF.* We are now assuming that $\ell': [0, k] \to \mathbb{R}^+$ is some concave and piecewise linear function on $[0, k]$ with $J + 1$ linear pieces, whose $i$th slope is $m_i$ and whose $i$th breakpoint is $B_i$. Let $\hat{\ell}$ be the CGF whose parameters are given in the lemma statement. By the proof of the other direction, we know that $\hat{\ell}$, like $\ell'$, is a nonnegative piecewise linear concave function on the interval $[0, k]$, with exactly $J + 1$ linear pieces. Since a piecewise linear function on $[0, k]$ is

uniquely determined by its breakpoints, and endpoints, and slopes, we simply need to check that $\ell'$ and $\hat{\ell}$ coincide at all of these.

The parameter choice $z_0 = \ell'(0)/k$ and $z_k = \ell'(k)/k$ guarantees that these functions match at inputs $0$ and $k$. From the proof of the first direction we know that the $i$th breakpoint of $\hat{\ell}$ will be $b_i = B_i$, so the functions match at breakpoints. It remains to check that their linear pieces have exactly the same slope.

Let $\ell_j = \ell'(b_j)$ for $j \in \{0\} \cup [J+1]$, where we again have used $b_0 = 0$ and $b_{J+1} = k$ for notational convenience. The $i$th linear piece of $\ell'$ has the slope

$$m_i = \frac{\ell_i - \ell_{i-1}}{b_i - b_{i-1}}.$$

From the proof of the first direction we know that the $i$th linear piece of $\hat{\ell}$, which is the linear piece corresponding to the interval $[b_{i-1}, b_i]$, has the slope

$$\hat{m}_i = z_k - z_0 - \sum_{j=1}^{J} a_j b_j + k \sum_{j=i}^{J} a_j = \frac{\ell_k}{k} - \frac{\ell_0}{k} - \sum_{j=1}^{J} a_j b_j + k \sum_{j=i}^{J} a_j.$$

We can simplify several terms using the fact that $a_j = \frac{1}{k}(m_j - m_{j+1})$. First of all,

$$k \sum_{j=i}^{J} a_j = \sum_{j=i}^{J} [m_j - m_{j+1}] = m_i - m_{J+1}.$$

Secondly,

$$k \sum_{j=1}^{J} a_j b_j = \sum_{j=1}^{J} (m_j - m_{j+1}) b_j = m_1 b_1 - m_{J+1} b_J + \sum_{j=2}^{J} m_j (b_j - b_{j-1})$$

$$= (\ell_1 - \ell_0) - m_{J+1} b_J + \sum_{j=2}^{J} [\ell_j - \ell_{j-1}]$$

$$= (\ell_1 - \ell_0) - m_{J+1} b_J + \ell_J - \ell_1$$

$$= \ell_J - \ell_0 - m_{J+1} b_J.$$

Therefore, the slope of the $i$th linear piece of $\hat{\ell}$ is

$$\hat{m}_i = \frac{\ell_{J+1}}{k} - \frac{\ell_0}{k} - \sum_{j=1}^{J} a_j b_j + k \sum_{j=i}^{J} a_j$$

$$= \frac{\ell_{J+1}}{k} - \frac{\ell_0}{k} - \frac{\ell_J}{k} + \frac{\ell_0}{k} + \frac{m_{J+1} b_J}{k} + m_i - m_{J+1}$$

$$= \frac{\ell_{J+1} - \ell_J}{k} + \frac{m_{J+1} b_J}{k} - m_{J+1} + m_i$$

$$= \frac{m_{J+1}(b_{J+1} - b_J)}{k} + \frac{m_{J+1} b_J}{k} - m_{J+1} + m_i$$

$$= \frac{m_{J+1}(k - b_J)}{k} + \frac{m_{J+1} b_J}{k} - m_{J+1} + m_i$$

$$= m_i.$$

Thus, the functions coincide. $\qquad\square$

## A.2 Finding the Optimal Piecewise Linear Approximation

Lemma 1 tells us that solving the sparse approximate reduction problem for a concave function $g$ is equivalent to finding a concave piecewise linear curve $\ell$ satisfying

$$g(i) \le \ell(i) \le (1 + \varepsilon) g(i) \text{ for all } i \in \{0, 1, 2, \ldots k\}, \tag{10}$$

---

**Algorithm 1** GREEDYPLCOVER$(g, \varepsilon)$

---

**Input**: $\varepsilon \geq 0$, concave function $g$
**Output**: piecewise linear $\ell$ with fewest linear pieces such that $g(i) \leq \ell(i) \leq (1 + \varepsilon)g(i)$
$\mathcal{L} \leftarrow \emptyset, u \leftarrow 0$    //u = smallest integer not covered by approximating line
**while** $u \leq k$ **do**
   $u', L \leftarrow$ NEXTLINE$(g, u, \varepsilon)$   //line covering widest range [u, u'-1]
   $\mathcal{L} \leftarrow \mathcal{L} \cup \{L\}, u \leftarrow u'$
**end while**
Return $\ell(x) = \min_{L \in \mathcal{L}} L(x)$

---

---

**Algorithm 2** NEXTLINE$(g, u, \varepsilon)$

---

**Input**: $\varepsilon \geq 0$, concave function $g$, integer $u \in \{0, 1, \ldots k\}$
**Output**: line $L$ satisfying $g(i) \leq L(i) \leq (1 + \varepsilon)g(i)$ for $i \in \{u, u + 1, \ldots, t\}$ for max integer $t$.
**if** $u \in \{k - 1, k\}$ **then**
   //If only 1 or 2 points to cover, this can be done with one line.
   Return $k + 1$, $L = $ LINETHROUGH$(\{k - 1, g(k - 1)\}, \{k, g(k)\})$
**end if**
$L \leftarrow$ LINETHROUGH$(\{u, (1 + \varepsilon)g(u)\}, \{u + 1, g(u + 1)\})$   //First candidate line
$u' = u + 2$
**while** $u' \leq k$ and $L(u') \leq (1 + \varepsilon)g(u')$ **do**
   **if** $L(u') < g(u')$ **then**
     //New candidate line, to ensure we return an upper bounding line
     $L \leftarrow$ LINETHROUGH$(\{u, (1 + \varepsilon)g(u)\}, \{u', g(u')\})$
   **end if**
   $u' = u' + 1$
**end while**
Return $u', L$

---

such that $\ell$ has a minimum number of linear pieces. This problem can be solved using Algorithm 1, which uses the function NEXTLINE (Algorithm 2) as a subroutine. For this pseudocode, LINETHROUGH is a conceptual subroutine that returns a line $L$ when given two points that define $L$. In practice this is implemented by storing those two points, or by storing one point and the line's slope.

**Theorem 2** *Algorithm 1 solves the sparsest approximate reduction problem in $O(k)$ operations.*

**Proof** To prove Algorithm 1 finds the optimal result, we must confirm three things. First, the linear pieces in $\mathcal{L}$ all upper bound the points $\{i, g(i)\}$. Second, for each $i \in \{0, 1, \ldots, k\}$, one of the lines satisfies $L$ satisfying $L(i) \leq (1 + \varepsilon)g(i)$. Third, there is no collection of lines $\mathcal{L}'$ of smaller cardinality satisfying these conditions.

*Greedy guarantee for Algorithm 2.* Each linear piece $L$ is found using Algorithm 2. For an integer $u$, this subroutine finds the line $L$ which satisfies

$$g(i) \leq L(i) \leq (1 + \varepsilon)g(i) \text{ for } i \in \{u, u + 1, \ldots, t\}. \tag{11}$$

for a maximum value of $t \leq k$. To see why, note that Algorithm 2 finds the line that goes through the point $\{u, (1 + \varepsilon)g(u)\}$ and the point $\{v, g(v)\}$ where $v > u$ is the smallest integer such that $L(v + 1) \geq g(v + 1)$. If does so by sequentially considering lines through the point $\{u, (1 + \varepsilon)g(u)\}$ and points $\{u', g(u')\}$ for increasing values of $u'$, updating the line at each step until it satisfies $L(u') = g(u')$ and $L(u' + 1) \geq g(u' + 1)$. By the concavity of $g$, this line will upper bound all remaining points $\{i, g(i)\}$. Also by concavity, any upper bounding line with a smaller slope would provide a worse approximation at $\{u, g(u)\}$, and therefore not provide the desired approximation at $u$. Meanwhile, any upper bounding line with a larger slope would have a worse approximation at every point $\{j, g(j)\}$ when $j > v$, so it may not satisfy (11) for the largest value of $t$. Thus, the line returned by Algorithm 2 provides an approximating line at $u$ with the farthest reach.

*Proof that greedy strategy is optimal.* Algorithm 1 uses Algorithm 2 to greedily grow a collection $\mathcal{L}$ of approximating lines. To see why this greedy strategy is optimal, consider any other collection

$\hat{\mathcal{L}}$ that provides a $(1 + \varepsilon)$ approximation everywhere, and we will show inductively that $|\mathcal{L}| \leq |\hat{\mathcal{L}}|$. At the beginning we assume we have not seen any of the lines from either collection. At each step, we require $\mathcal{L}$ and $\hat{\mathcal{L}}$ to produce a line that covers the smallest integer for which they previously did not provide a cover. In the first step, $\mathcal{L}$ must provide a line that covers $u_1 = 0$ and $\hat{\mathcal{L}}$ must also provide a line that covers $\hat{u}_1 = 0$. After the first step, $\mathcal{L}$ produces a line covering $\{u_1 = 0, 1, \ldots, t\}$ where $t$, and $\hat{\mathcal{L}}$ produces a line covering $\{\hat{u}_1 = 0, 1, \ldots, \hat{t}\}$. Because Algorithm 2 is guaranteed to find a line with a maximum reach, we have $\hat{t} \leq t$, and we begin the next step with $u_2 = t + 1$ and $\hat{u}_2 = \hat{t} + 1 < t + 1$. Assume inductively that at the $i$th step, $\mathcal{L}$ must provide a line covering an integer $u_i$ and $\hat{\mathcal{L}}$ must provide a line covering $\hat{u}_i$, where $u_i \geq \hat{u}_i$. After this step, it is impossible for $\hat{\mathcal{L}}$ to surpass $\mathcal{L}$ in such a way that $\hat{u}_{i+1} > u_{i+1}$. This would imply that $\hat{\mathcal{L}}$ found a line that provides a $(1 + \varepsilon)$-approximation for the entire range $\{\hat{u}_i, \hat{u}_i + 1, \ldots, \hat{u}_{i+1}\}$ for $\hat{u}_i \leq u_i$ and $\hat{u}_{i+1} > u_{i+1}$, contradicting the fact that $\mathcal{L}$ finds the line that covers the set $\{u_i, u_i + 1, \ldots, u_{i+1}\}$ for the largest value of $u_{i+1}$. We see therefore that $u_i \geq \hat{u}_i$ at every step, so it is impossible for $\hat{\mathcal{L}}$ to produce a set of lines providing the desired approximation for the integers $\{0, 1, 2, \ldots k\}$ before $\mathcal{L}$ does.

*Runtime guarantee.* Regarding runtime, Algorithms 1 and 2 together visit each integer $i \in \{0, 1, 2, \ldots, k\}$ in turn once and performs a constant number of operations to check whether a certain line provides a desired approximation to the point $\{i, g(i)\}$, and possibly to define a new line if the approximation is not satisfied. Computing and storing the information needed to define a line through two points is easily done in a few operations, so the overall work to find and store all of $\mathcal{L}$ is $O(k)$. Lines in $\mathcal{L}$ will already be sorted in terms of the value of their slopes, so in $O(|\mathcal{L}|) \leq O(k)$ time it is simple to compute intersection points of consecutive lines, and find the slopes and breakpoints needed to fully define the piecewise linear function $\ell(x) = \min_{L \in \mathcal{L}} L(x)$. $\square$

## A.3 Bounds on optimal reduction size

We now provide several bounds on the number of linear pieces needed to solve SpAR for different concave functions $g$ over an interval $[0, k]$. These can be viewed as stand-alone results about approximating concave functions at integer points with piecewise linear curves. Given the equivalence in Lemma 1, each of these results immediately implies a bound on number of CB-gadgets needed to approximately model a concave cardinality function $f_e(A) = g(|A|)$.

**Theorem 3** *Let $g \colon [0, k] \to \mathbb{R}^+$ be concave and let $\varepsilon \geq 0$. Algorithm 1 will return a piecewise linear function $\ell$ with at most $\min\{1 + \lfloor k/2 \rfloor, 2 + 2\lceil \log_{1+\varepsilon} k \rceil\}$ linear pieces satisfying $g(i) \leq \ell(i) \leq (1 + \varepsilon)g(i)$ for $i \in \{0, 1, 2, \ldots k\}$. As $\varepsilon \to 0$, $\log_{1+\varepsilon} k$ behaves as $\frac{1}{\varepsilon} \log k$.*

**Proof** Let $q$ be the piecewise linear curve obtained by performing linear interpolation on the points $\{i, g(i)\}$ for $i \in \{0, 1, 2, \ldots, k\}$. This has $k$ linear pieces and exactly matches $g$ at integer points, so we know we immediately have an upper bound of $k$ linear pieces. In order to prove the logarithmic upper bound, we will bound the number of linear pieces needed to approximate $q$ on the entire interval $[1, k]$ by taking a subset of its linear pieces. Our argument follows similar previous results for approximating a concave function with a logarithmic number of linear pieces [7].

We will prove the result first under the assumption that $g$ (and therefore $q$) is a monotonically increasing concave function. For any value $y \in [1, k]$, not necessarily an integer, $q(y)$ lies on a line which we will denote by $q^{(y)}(x) = M_y \cdot x + B_y$, where $M_y \geq 0$ is the slope and $B_y \geq 0$ is the intercept. When $y = i$ is an integer, it may be the breakpoint between two distinct linear pieces, in which case we use the rightmost line so that $q^{(y)} = q^{(i)}$, where $q^{(i)}(x) = M_i \cdot x + B_i$ has slope $M_i = g(i + 1) - g(i)$ and $B_i = g(i) - M_i \cdot i$. For any $z \in (y, k)$, the line $q^{(y)}$ provides a $z/y$ approximation to $q(z) = q^{(z)}(z)$, since

$$q^{(y)}(z) = M_y \cdot z + B_y \leq \frac{z}{y}(M_y \cdot y + B_y) = \frac{z}{y}q(y) \leq \frac{z}{y}q(z).$$

Equivalently, the line $q^{(y)}$ provides a $(1+\varepsilon)$-approximation for every $z \in [y, (1+\varepsilon)y]$. Thus, it takes $J$ linear pieces to cover the set of intervals $[1, (1 + \varepsilon)], [(1 + \varepsilon), (1 + \varepsilon)^2], \ldots, [(1 + \varepsilon)^{J-1}, (1 + \varepsilon)^J]$ for a positive integer $J$, and overall at most $1 + \lceil \log_{1+\varepsilon} k \rceil$ linear pieces to cover all of $[0, k]$.

If $g$ and $q$ instead are monotonically decreasing, then we can apply the above procedure to the function $\hat{q}(x) = q(k - x)$. This is a monotonically increasing mirror image of $q$, and selecting an

approximating subset of linear pieces of $\hat{q}$ is equivalent to selecting an approximating subset of linear pieces of $q$. If $g$ is not monotonic and not the constant function $g(x) = 0$, then as a concave function, the linear interpolation $q$ will monotonically increase on an interval $[0, r]$ for some integer $r < k$, and then monotonically decrease on the interval $[r, k]$. We can apply the same procedures to find an approximating cover for $[0, r]$ and then another approximating cover for $[r, k]$, and then combine the two, for a total of $2 + 2\lceil \log_{1+\varepsilon} k \rceil$ linear pieces. Since Algorithm 1 finds a minimum sized set of linear pieces to cover $g$ at integer points, it must also have at most this many linear pieces.

Finally, when $\varepsilon$ is very small, we can do slightly better than use all $k$ linear pieces that define $q$. We can always produce a piecewise linear function matching $g$ exactly at integer values by joining consecutive *disjoint* pairs of points by a line, e.g., join $\{0, g(0)\}$ and $\{1, g(1)\}$ with a line, then join $\{2, g(2)\}$ and $\{3, g(3)\}$ with a line, etc. This leads to a lower bound of $\lfloor k/2 \rfloor + 1$ lines needed for any positive integer $k$. For $\varepsilon = 0$, Algorithm 1 will this set of linear pieces. $\qquad \square$

**Lower bound for the square root function** In order to prove a lower bound result, we will use the following lemma of Magnanti and Stratila [7] on the number of linear pieces needed to approximate the square root function.

**Lemma A.1** *(Lemma 3 in [7]) Let $\varepsilon > 0$ and $\phi(x) = \sqrt{x}$. Let $\psi$ be a piecewise linear function whose linear pieces are all tangent lines to $\phi$, satisfying $\psi(x) \leq (1+\varepsilon)\phi(x)$ for all $x \in [l, u]$ for $0 < l < u$. Then $\psi$ contains at least $\lceil \log_{\gamma(\varepsilon)} \frac{u}{l} \rceil$ linear pieces, where $\gamma(\varepsilon) = (1+2\varepsilon(2+\varepsilon)+2(1+\varepsilon)\sqrt{\varepsilon(2+\varepsilon)})^2$. There exists a piecewise linear function $\psi^*$ of this form with exactly $\lceil \log_{\gamma(\varepsilon)} \frac{u}{l} \rceil$ linear pieces.[1] As $\varepsilon \to 0$, this values behaves as $\frac{1}{\sqrt{32\varepsilon}} \log \frac{u}{l}$.*

This result is concerned with approximating the square root function for *all* values on a *continuous* interval. Therefore, it does not imply any bounds on approximating a discrete set of points of a concave function. In fact, because we can always cover $k$ points with $k$ linear pieces, we know that for any function $f(k)$ of $k$, there is no lower bound of the form $\tau(\varepsilon)f(k)$ that holds for *all* $\varepsilon > 0$, if $\tau$ is a function such that $\tau(\varepsilon) \to \infty$ as $\varepsilon \to 0$. The best we can expect is a lower bound that holds for $\varepsilon$ values that may still go to zero as $k \to \infty$, but are bounded in such a way that we do not contradict the $O(k)$ upper bound. We prove such a result for the square root function using Lemma A.1 as a black box. When $\varepsilon$ falls below the bound we use in the following theorem statement, forming $O(k)$ linear pieces will be nearly optimal.

**Theorem 4** *Let $g(x) = \sqrt{x}$ and $\varepsilon \geq k^{-\delta}$ for a constant $\delta \in (0, 2)$. Every piecewise linear $\ell$ satisfying $g(i) \leq \ell(i) \leq (1+\varepsilon)g(i)$ for $i \in \{0, 1, \ldots k\}$ contains $\Omega(\log_{\gamma(\varepsilon)} k)$ linear pieces where $\gamma(\varepsilon) = (1 + 2\varepsilon(2+\varepsilon) + 2(1+\varepsilon)\sqrt{\varepsilon(2+\varepsilon)})^2$. This behaves as $\Omega(\varepsilon^{-1/2} \log k)$ as $\varepsilon \to 0$.*

**Proof** Let $\mathcal{L}^*$ be the optimal set of linear pieces returned by running Algorithm 1 for the function $g(x) = \sqrt{x}$. In order to show $|\mathcal{L}^*| = \Omega(\log_{\gamma(\varepsilon)} k)$, we will construct a new set of linear pieces $\mathcal{L}$ that has asymptotically the same number of linear pieces as $\mathcal{L}^*$, but also provides a $(1+\varepsilon)$-approximation for all $x$ in an interval $[k^\beta, k]$ for some constant $\beta < 1$. Invoking Lemma A.1 will imply a lower bound on the size of $\mathcal{L}$, and in turn the number of linear pieces in $\mathcal{L}^*$.

Observe that $\mathcal{L}^*$ will include the line going through $\{0, g(0)\}$ and $\{1, g(1)\}$, and may include the line that goes through points $\{k-1, g(k-1)\}$ and $\{k, g(k)\}$ depending on $\varepsilon$ and $k$. Otherwise, all of the lines it includes go through exactly one point $\{i, g(i)\}$ for some integer $i$. All of these lines bound $g(x) = \sqrt{x}$ above at *integer* points, but they may cross below $g$ at non-integer values of $x$. To apply Lemma A.1, we would like to obtain a set of linear pieces that are all tangent lines to $g$. We accomplish this by replacing each linear piece with linear pieces that are tangent to $g$ at some point. For a positive integer $j$, let $L_j$ denote the line tangent to $g(x) = \sqrt{x}$ at $x = j$, which is given by

$$L_j(x) = \frac{1}{2\sqrt{j}}(x - j) + \sqrt{j}. \qquad (12)$$

We form a new set of linear pieces $\mathcal{L}$ made up of lines tangent to $g$ using the following replacements:

---

[1]This additional statement is not included explicitly in the statement of Lemma 3 in [7], but it follows directly from the proof of the lemma, which shows how to construct such an optimal function $\psi^*$.

- Replace the line going through $\{0, g(0)\}$ and $\{1, g(1)\}$ with $L_1$.
- If $\mathcal{L}^*$ includes the line through $\{k - 1, g(k - 1)\}$ and $\{k, g(k)\}$, replace it with $L_{k-1}$ and $L_k$
- For a line crossing through a point $\{i, g(i)\}$ for some integer $i \in [2, k - 1]$, replace the line with with $L_{j-1}$, $L_j$, and $L_{j+1}$.

By the concavity of $g$, this replacement can only improve the approximation guarantee at integer points. Therefore, $\mathcal{L}$ provides a $(1 + \varepsilon)$-approximation at integer values, is made up strictly of lines that are tangent to $g$, and contains at most three times the number of lines in $\mathcal{L}^*$.

The concavity of $g$ also tells us that if a single line $L \in \mathcal{L}$ provides a $(1 + \varepsilon)$-approximation at consecutive integers $i$ and $i+1$, then $L$ provides the same approximation guarantee for all $x \in [i, i+1]$. However, if two integers $i$ and $i + 1$ are not *both* covered by the *same* line in $\mathcal{L}$, then this does not apply and we cannot guarantee $\mathcal{L}$ provides a $(1+\varepsilon)$-approximation for every $x \in [i, i+1]$. There can be at most $|\mathcal{L}|$ intervals of this form, since these define intersection points between two consecutive approximating linear pieces in $\mathcal{L}$.

By Lemma A.1, we can cover an entire interval $[i, i + 1]$ for any integer $i$ using a set of $\lceil \log_{\gamma(\varepsilon)} \left(1 + \frac{1}{i}\right) \rceil$ linear pieces that are tangent to $g$ somewhere in $[i, i + 1]$. Since $1 + \sqrt{\varepsilon} \le \gamma(\varepsilon)$, it in fact takes only one linear piece to cover $[i, i + 1]$ as long as $i \ge 1/\sqrt{\varepsilon}$, since then we have $1 + 1/i \le 1 + \sqrt{\varepsilon}$ and therefore $\log_{\gamma(\varepsilon)}(1 + 1/i) \le \log_{\gamma(\varepsilon)}(1 + \sqrt{\varepsilon}) \le 1$. Since $\varepsilon \ge k^{-\delta}$, interval $[i, i + 1]$ can be covered by a single linear piece if $i \ge k^{\delta/2}$. Therefore, for each interval $[i, i + 1]$, with $i \ge k^{\delta/2}$, that is not already covered by a single linear piece in $\mathcal{L}$, we add one more linear piece to $\mathcal{L}$ to cover this interval. This at most doubles the size of $\mathcal{L}$.

The resulting set $\mathcal{L}$ will have at most 6 times as many linear pieces as $\mathcal{L}^*$, and is guaranteed to provide a $(1 + \varepsilon)$-approximation for all integers, as well as the entire continuous interval $[k^{\delta/2}, k]$. Since $\delta$ is a fixed constant strictly less than 2, applying Lemma A.1 shows that $\mathcal{L}$ has at least

$$\left\lceil \log_{\gamma(\varepsilon)} \frac{k}{k^{\delta/2}} \right\rceil = \Omega(\log_{\gamma(\varepsilon)} k^{1-\delta/2}) = \Omega(\log_{\gamma(\varepsilon)} k)$$

linear pieces. Therefore, $|\mathcal{L}^*| = \Omega(\log_{\gamma(\varepsilon)} k)$ as well. $\qquad\square$

### A.4 Improved Bound for the Clique Function

When approximating the function $g(x) = x(k - x)$ for an integer $k$, Algorithm 1 will in fact find a piecewise linear curve with at most $O(\varepsilon^{-1/2} \log \log \frac{1}{\varepsilon})$ linear pieces. We prove this by highlighting a different approach for constructing a piecewise linear curve with this many linear pieces, which upper bounds the minimum number of linear pieces returned by Algorithm 1. We refer to this as the clique function, since the submodular function $f_e$ defined by $f_e(A) = g(|A|) = |A|(|e| - |A|)$ is the cut function for a complete graph (i.e., a clique) on a set of $k = |e|$ nodes.

As we did with Algorithm 1, we want to build a set of linear pieces $\mathcal{L}$ that provides and upper bounding $(1 + \varepsilon)$-cover of $g$ at integer values in $[0, k]$. We start by adding the line $g^{(0)}(x) = (g(1) - g(0))x + g(0) = (k - 1) \cdot x$ to $\mathcal{L}$, which perfectly covers the first two points $\{0, g(0)\}$ and $\{1, g(1)\}$. In the remainder of the procedure we will find a set of linear pieces to $(1 + \varepsilon)$-cover $g$ at *every* value of $x \in [1, k/2]$, even non-integer $x$. The fact that the function satisfies $g(x) = g(k - x)$ implies that we can double the number of linear pieces to ensure we also cover the interval $[k/2, k]$.

We apply a greedy procedure similar to Algorithm 1, summarized in Algorithm 3. At each iteration we consider a leftmost endpoint $z_i$ which is the largest value in $[1, k/2]$ for which we already have a $(1 + \varepsilon)$-approximation. In the first iteration, we have $z_1 = 1$. We then would like to find a new linear piece that provides a $(1 + \varepsilon)$-approximation for all values from $z_i$ to some $z_{i+1}$, where the value of $z_{i+1}$ is maximized. We restrict to linear pieces that are tangent to $g$. The line tangent to $g$ at $t \in [1, k/2]$ is given by

$$g_t(x) = kx - 2tx + t^2 \,. \tag{13}$$

We find $z_{i+1}$ in two steps:

1. **Step 1:** Find the maximum value $t$ such that $g_t(z_i) = (1 + \varepsilon)g(z_i)$.

**Algorithm 3** Find a $(1 + \varepsilon)$-cover for the clique function.

---

**Input:** Integer $k$, $\varepsilon \geq 0$
**Output:** $(1 + \varepsilon)$ cover for the clique function $g(x) = x(k - x)$ for $[1, k/2]$.
$\mathcal{L} = \{g^{(0)}\}$, where $g^{(0)}(x) = (k-1)x$
$z = 1$
**do**
$\quad t \leftarrow z + \sqrt{z(k - z)\varepsilon}$
$\quad z \leftarrow \frac{t}{1+\varepsilon} + \frac{k\varepsilon}{2(1+\varepsilon)} + \frac{1}{2(1+\varepsilon)} \left(k^2\varepsilon^2 + 4\varepsilon t(k - t)\right)^{1/2}$
$\quad \mathcal{L} \leftarrow \mathcal{L} \cup \{g_t\}$, where $g_t(x) = kx - 2tx + t^2$
**while** $z_{i+1} < k/2$
Return $\ell$ defined by $\ell(x) = \min_{L \in \mathcal{L}} L(x)$

---

2. **Step 2:** Given $t$, find the maximum $z_{i+1}$ such that $g_t(z_{i+1}) = (1 + \varepsilon)g(z_{i+1})$.

After completing these two steps, we add the linear piece $g_t$ to $\mathcal{L}$, knowing that it covers all values in $[z_i, z_{i+1}]$ with a $(1 + \varepsilon)$-approximation. At this point, we will have a cover for all values in $[0, z_{i+1}]$, and we begin a new iteration with $z_{i+1}$ being the largest value covered. We continue until we have covered all values up until $z_{i+1} \geq k/2$.

**Lemma A.2** *For any $z_i \in [1, k/2]$, the values of $t$ and $z_{i+1}$ given in steps 1 and 2 are given by*

$$t = z_i + \sqrt{z_i(k - z_i)\varepsilon} \tag{14}$$

$$z_{i+1} = \frac{t}{1 + \varepsilon} + \frac{k\varepsilon}{2(1 + \varepsilon)} + \frac{1}{2(1 + \varepsilon)} \left(k^2\varepsilon^2 + 4\varepsilon t(k - t)\right)^{1/2} \tag{15}$$

**Proof** The proof simply requires solving two different quadratic equations. For Step 1:

$$g_t(z_i) = (1 + \varepsilon)g(z_i) \iff kz_i - 2tz_i + t^2 = (1 + \varepsilon)(z_i k - z_i^2)$$
$$\iff t^2 - 2z_i t - \varepsilon z_i k + (1 + \varepsilon)z_i^2 = 0$$

Taking the larger solution to maximize $t$:

$$t = \frac{1}{2}\left(2z_i + \sqrt{4z_i^2 - 4(1 + \varepsilon)z_i^2 + 4\varepsilon k z_i}\right) = z_i + \sqrt{z_i(k - z_i)\varepsilon}.$$

For Step 2:

$$g_t(z_{i+1}) = (1 + \varepsilon)g(z_{i+1}) \iff kz_{i+1} - 2tz_{i+1} + t^2 = (1 + \varepsilon)(z_{i+1}k - z_{i+1}^2)$$
$$\iff (1 + \varepsilon)z_{i+1}^2 + z_{i+1}(-\varepsilon k - 2t) + t^2 = 0.$$

We again take the larger solution to this quadratic equation since we want to maximize $z_{i+1}$:

$$z_{i+1} = \frac{1}{2(1 + \varepsilon)}\left(\varepsilon k + 2t + \sqrt{\varepsilon^2 k^2 + 4t\varepsilon k + 4t^2 - 4(1 + \varepsilon)t^2}\right)$$
$$= \frac{1}{2(1 + \varepsilon)}\left(\varepsilon k + 2t + \sqrt{\varepsilon^2 k^2 + 4t\varepsilon(k - t)}\right).$$

$\square$

Since $z_1 = 1$, if $\varepsilon \geq 1$, then

$$z_2 \geq \frac{1}{2(1 + \varepsilon)}(2k\varepsilon) = \frac{k\varepsilon}{1 + \varepsilon} \geq \frac{k}{2},$$

so after one step we have covered the entire interval $[1, k/2]$. We can therefore focus on $\varepsilon < 1$. We are now ready to prove the result given in the main text.

**Theorem 5** *Let $g(x) = x \cdot (k - x)$ for a positive integer $k$. For $\varepsilon > 0$, the approximating function $\ell$ returned by Algorithm 1 will have $O(\varepsilon^{-1/2} \log \log \varepsilon^{-1})$ linear pieces.*

**Proof** The result holds if we can show that Algorithm 3 outputs a collection $\mathcal{L}$ with at most $O(\varepsilon^{-1/2} \log \log \frac{1}{\varepsilon})$ lines for any $\varepsilon < 1$. We get a loose bound for the value of $t$ in Lemma A.2 by noting that $(k - z_i) \geq k/2 \geq z_i$:

$$t = z_i + \sqrt{z_i \varepsilon (k - z_i)} \geq z_i + \sqrt{z_i^2 \varepsilon} = z_i(1 + \sqrt{\varepsilon}). \tag{16}$$

Since we assumed $\varepsilon < 1$, we know that

$$\frac{t}{1 + \varepsilon} \geq \frac{z_i(1 + \sqrt{\varepsilon})}{1 + \varepsilon} > z_i. \tag{17}$$

Therefore, from (15) we see that

$$z_{i+1} > z_i + \frac{k\varepsilon}{2(1 + \varepsilon)} + \frac{1}{2(1 + \varepsilon)} \left(k^2 \varepsilon^2 + 4\varepsilon t(k - t)\right)^{1/2} \tag{18}$$

$$> z_i + \frac{k\varepsilon}{2(1 + \varepsilon)} + \frac{1}{2(1 + \varepsilon)} \left(k^2 \varepsilon^2\right)^{1/2} = z_i + \frac{k\varepsilon}{1 + \varepsilon}. \tag{19}$$

From this we see that at each iteration, we cover an additional interval of length $z_{i+1} - z_i > \frac{k\varepsilon}{1+\varepsilon}$, and therefore we know it will take at most $O(1/\varepsilon)$ iterations to cover all of $[1, k/2]$. This upper bound is loose, however. The value of $z_{i+1} - z_i$ in fact increases significantly with each iteration, allowing the algorithm to cover larger and larger intervals as it progresses.

Since $z_1 = 1$ and $z_{i+1} - z_i \geq \frac{k\varepsilon}{1+\varepsilon}$, we see that $z_j \geq k\varepsilon$ for all $j \geq 3$. For the remainder of the proof, we focus on bounding the number of iterations it takes to cover the interval $[k\varepsilon, k/2]$. We separate the progress made by Algorithm 3 into different rounds. Round $j$ refers to the set of iterations that the algorithm spends to cover the interval

$$R_j = \left[k\varepsilon^{\left(\frac{1}{2}\right)^{j-1}}, k\varepsilon^{\left(\frac{1}{2}\right)^j}\right], \tag{20}$$

For example, Round 1 starts with the iteration $i$ such that $z_i \geq k\varepsilon$, and terminates when the algorithm reaches an iteration $i'$ where $z_{i'} \geq k\varepsilon^{1/2}$. A key observation is that it takes less than $4/\sqrt{\varepsilon}$ iterations for the algorithm to finish Round $j$ for any value of $j$. To see why, observe that from the bound in (18) we have

$$z_{i+1} - z_i > \frac{k\varepsilon}{2(1 + \varepsilon)} + \frac{1}{2(1 + \varepsilon)} \left(k^2 \varepsilon^2 + 4\varepsilon t(k - t)\right)^{1/2}$$

$$> \frac{1}{2(1 + \varepsilon)} \left(4\varepsilon t(k - t)\right)^{1/2}$$

$$\geq \frac{1}{2(1 + \varepsilon)} \left(4\varepsilon z_i \frac{k}{2}\right)^{1/2}$$

$$> \frac{\sqrt{2}}{2} \frac{\sqrt{k\varepsilon}}{(1 + \varepsilon)} \sqrt{z_i}.$$

For each iteration $i$ in Round $j$, we know that $z_i \geq k\varepsilon^{\left(\frac{1}{2}\right)^{j-1}}$, so that

$$z_{i+1} - z_i > \frac{\sqrt{2}}{2} \frac{\sqrt{k\varepsilon}}{(1 + \varepsilon)} \sqrt{k\varepsilon^{\left(\frac{1}{2}\right)^{j-1}}} \geq \frac{\sqrt{2}}{2} \frac{k\varepsilon^{\frac{1}{2} + \left(\frac{1}{2}\right)^j}}{1 + \varepsilon} = C \cdot k \cdot \varepsilon^{\frac{1}{2} + \left(\frac{1}{2}\right)^j}, \tag{21}$$

where $C = \sqrt{2}/(2(1 + \varepsilon))$ is a constant larger than $1/4$. Since each iteration of Round $j$ covers an interval of length at least $C \cdot k \cdot \varepsilon^{\frac{1}{2} + \left(\frac{1}{2}\right)^j}$, and the right endpoint for Round $j$ is $k\varepsilon^{\left(\frac{1}{2}\right)^j}$, the maximum number of iterations needed to complete Round $j$ is

$$\frac{k\varepsilon^{\left(\frac{1}{2}\right)^j}}{C \cdot k \cdot \varepsilon^{\frac{1}{2} + \left(\frac{1}{2}\right)^j}} = \frac{1}{C\sqrt{\varepsilon}}. \tag{22}$$

Therefore, after $p$ rounds, the algorithm will have performed $O(p \cdot \varepsilon^{-1/2})$ iterations, to cover the interval $[1, k\varepsilon^{\left(\frac{1}{2}\right)^p}]$. Since we set out to cover the interval $[1, k/2]$, this will be accomplished as soon

**Algorithm 4** SPARSECARD$(f, \varepsilon)$

---

**Input**: $\varepsilon \geq 0$, function $f(S) = \sum_{e \in E} f_e(S \cap e) = \sum_{e \in E} g_e(|S \cap e|)$ on ground set $V$
**Output**: Set $S' \subseteq V$ satisfying $f(S') \leq (1 + \varepsilon) \min_{S \subseteq V} f(S)$.
$\mathcal{A} \leftarrow \emptyset, \mathcal{E} \leftarrow \emptyset$ //initialize auxiliary node and edge set for reduced graph

**for** $e \in E$ **do**
    //Step 1: solve piecewise linear function approximation problem
    $\ell_e \leftarrow$ GREEDYPLCOVER$(g_e, \varepsilon)$

    //Step 2: Construct graph gadget for e with auxiliary nodes $\mathcal{A}_e$
    $G_e = (e \cup \mathcal{A}_e \cup \{s, t\}, \mathcal{E}_e) \leftarrow$ CGFTOGADGET$(\ell_e)$

    //Step 3: Add new auxiliary nodes and edges for building graph G
    $\mathcal{A} \leftarrow \mathcal{A} \cup \mathcal{A}_e, \mathcal{E} \leftarrow \mathcal{E} \cup \mathcal{E}_e$

**end for**
$G = (V \cup \mathcal{A} \cup \{s, t\}, \mathcal{E})$       //Build graph G modeling f
$T = $ MINSTCUT$(G)$           //Find minimum s-t cut
Return $S' = T \cap V$         //Ignore auxiliary nodes

---

as $p$ satisfies $\varepsilon^{\left(\frac{1}{2}\right)^p} \geq 1/2$, which holds as long as $p \geq \log_2 \log_2 \frac{1}{\varepsilon}$:

$$\varepsilon^{\left(\frac{1}{2}\right)^p} \geq 1/2 \iff \left(\frac{1}{2}\right)^p \log_2 \varepsilon \geq -1$$

$$\iff \log_2 \varepsilon \geq -2^p$$

$$\iff \log_2 \frac{1}{\varepsilon} \leq 2^p$$

$$\iff \log_2 \log_2 \frac{1}{\varepsilon} \leq p.$$

This means that the number of iteration of Algorithm 3, and therefore the number of linear pieces in $\mathcal{L}$, is bounded above by $O(\varepsilon^{-1/2} \log \log \frac{1}{\varepsilon})$. $\qquad \square$

### A.5 Theoretical Guarantees for SPARSECARD

Algorithm 2 gives pseudocode for SPARSECARD, which relies on Algorithm 1 for finding a piecewise linear approximation $\ell_e$ for each function $g_e$, and Algorithm 5 for converting the piecewise linear function into a combination of CB-gadgets that approximately models $f_e(A) = g_e(|A|)$.

**Theorem 6** *Let $n = |V|$ and $R = |E|$. When $\varepsilon = 0$, the graph constructed by SPARSECARD will have $O(\sum_{e \in E} |e|)$ nodes and $O(\sum_{e \in E} |e|^2)$ edges. When $\varepsilon > 0$, the graph will have $O(n + \sum_{e \in E} \varepsilon^{-1} \log |e|)$ nodes and $O(\varepsilon^{-1} \sum_{e \in E} |e| \log |e|)$ edges. In either case, and the method will return a set $T$ satisfying $f(T) \leq (1 + \varepsilon) \min_{S \subseteq V} f(S)$.*

**Proof** Each auxiliary node in the graph construction for SPARSECARD is unique to its own CB-gadget, but the source node $s$ and sink node $t$ are shared across all combined gadgets, and each node $v \in V$ will show up in multiple CB-gadgets. When $\varepsilon = 0$, we will need $\lfloor |e|/2 \rfloor + 1$ CB-gadgets to exactly model the function $f_e(A) = g_e(|A|)$ for a node set $e$. This comes from Theorem 3 and the fact that we need $\lfloor k/2 \rfloor + 1$ linear pieces to exactly match the function $g_e$ at integer points $\{0, 1, 2, \ldots, k = |e|\}$. Overall then, the total number of auxiliary nodes is

$$|\mathcal{A}| = \sum_{e \in E} (\lfloor |e|/2 \rfloor + 1),$$

and when we include $\{s, t\}$ and $V$, the total number of nodes is asymptotically still $O(\sum_{e \in E} |e|)$. Since each CB-gadget for $e$ requires $2|e|$ directed edges, the number of edges due to CB-gadgets is

$$\sum_{e \in E} 2|e| \cdot (\lfloor |e|/2 \rfloor + 1) = O(\sum_{e \in E} |e|^2).$$

**Algorithm 5** CGFTOGADGET($\ell_e$)

---

**Input**: Piecewise linear function $\ell_e$ on $[0, k]$ where $k = |e|$ for a set $e$
**Output**: $G_e = (e \cup \mathcal{A}_e \cup \{s, t\}, \mathcal{E}_e)$: combination of CB-gadgets modeling $\hat{f}_e(A) = \ell_e(|A|)$.

```
// Extract information about linear pieces
```
$J = ($ # of linear pieces of $\ell_e) - 1$
$\{m_1, m_2, \ldots, m_{J+1}\} = $ slopes of linear pieces of $\ell_e$
$\{b_1, b_2, \ldots, b_J\} = $ breakpoints of $\ell_e$
$z_0 = \ell_e(0)/k$
$z_k = \ell_e(k)/k$

```
// Build collection of CB-gadgets
```
$\mathcal{A}_e \leftarrow \emptyset, \mathcal{E}_e \leftarrow \emptyset$
$s \leftarrow$ source node
$t \leftarrow$ sink node
**for** $v \in e$ **do**
    Add directed edge $(s, v)$ of weight $z_0$ to $\mathcal{E}_e$
    Add directed edge $(v, t)$ of weight $z_k$ to $\mathcal{E}_e$
    **for** $i = 1$ to $J$ **do**
        ```// Edges for the i-th CB-gadget gadget```
        $a_i \leftarrow \frac{1}{k}(m_i - m_{i+1})$
        $\mathcal{A}_e \leftarrow \mathcal{A}_e \cup \{v_{e,i}\}$
        Add directed edge $(v, v_{e,i})$ of weight $a_i(k - b_i)$ to $\mathcal{E}_e$
        Add directed edge $(v_{e,i}, v)$ of weight $a_i b_i$ to $\mathcal{E}_e$
    **end for**
**end for**
Return $G_e = (e \cup \mathcal{A}_e \cup \{s, t\}, \mathcal{E}_e)$

---

Adding in the edges between $V$ and $\{s, t\}$ increases the total number of edges by $O(n)$. The bound on the number of nodes and edges when $\varepsilon > 0$ follows the same steps, except we apply that fact that each $f_e$ is approximately modeled using a set of $O(\frac{1}{\varepsilon} \log |e|)$ CB-gadgets.

To prove the approximation result, let $G_e = (\{s, t\} \cup e \cup \mathcal{A}_e)$ be the small graph constructed in Algorithm 5 via a combination of CB-gadgets to approximately model $f_e(A) = g_e(|A|)$. By design, for every $A \subseteq e$ the cut properties of this small graph satisfy

$$\min_{B \subseteq \mathcal{A}_e} \mathbf{cut}_{G_e}(\{s\} \cup B \cup A) = \ell_e(|A|). \tag{23}$$

where $\ell_e$ is the piecewise linear function constructed by Algorithm 1 to approximate $g_e$, and $\mathbf{cut}_{G_e}$ is the cut function of the graph $G_e$. In other words, if you take any subset of the nodes in $e$ and rearrange auxiliary nodes $\mathcal{A}_e$ so that you get the minimum cut penalty subject to $A$ being on the source side, that penalty is $\ell_e(|A|) \leq (1 + \varepsilon)g_e(|A|) = (1 + \varepsilon)f_e(A)$. The minimum $s$-$t$ cut set in the overall graph $G$ is therefore

$$T = \operatorname{argmin}_{S \subseteq V} \left\{ \sum_{e \in E} \min_{B \subseteq \mathcal{A}_e} \mathbf{cut}_{G_e}(\{s\} \cup B \cup (e \cap S)) \right\} = \operatorname{argmin}_{S \subseteq V} \left\{ \sum_{e \in E} \ell_e(|e \cap S|) \right\}.$$

If $S^* = \operatorname{argmin}_{S \subseteq V} f(S) = \sum_{e \in E} f_e(S \cap e)$ is the minimizer of the original decomposable submodular function, then we have

$$f(S^*) \leq f(T) = \sum_{e \in E} \ell_e(|e \cap T|) \leq (1+\varepsilon) \sum_{e \in E} g_e(|e \cap T|) = (1+\varepsilon) \sum_{e \in E} f_e(e \cap T) = (1+\varepsilon)f(T).$$

$\square$

# B Improved Approach for Symmetric Functions

In many applications of decomposable submodular function minimization, some or all of the component functions $f_e$ are symmetric, meaning that for every $A \subseteq e$ they satisfy $f_e(A) = f_e(e \backslash A)$.

Another common assumption is that $f_e(e) = f_e(\emptyset) = 0$. These two conditions are typically assumed in applications where $f_e$ represents a generalized cut penalty for a hyperedge $e$ in some hypergraph [9, 4, 6, 10, 2]. Symmetry implies that if a hyperedge is partitioned into sets $A$ and $e \backslash A$, it does not matter which side of the cut each set is on. The condition $f_e(e) = f_e(\emptyset) = 0$ reflects the fact that there should be no cut penalty if a hyperedge $e$ is completely contained on one side of the cut other the other.

Assuming these two conditions leads simply to a special case of the more general functions we considered above, which can therefore be handled using the reduction technique outlined previously. However, there is a slightly more efficient graph reduction technique for symmetric functions that requires roughly half the number of edges. While this makes no different for our asymptotic results, cutting down the number of edges in the reduced graph by a factor of two can be very worthwhile in practice. We therefore provide details for this more efficient reduction strategy. In addition to the symmetric property, we will assume that all functions $f_e$ we consider satisfy $f_e(e) = f_e(\emptyset) = 0$. Slight adjustments can be made to also handle the case where the function is not symmetric but still satisfies $f_e(e) = f_e(\emptyset) > 0$. Note that the reduction here is what we use when modeling the clique submodular function $f_e(A) = |A||e\backslash A|$ that arises frequently in hypergraph clustering applications and image segmentation.

## B.1 The symmetric cardinality-based gadget

The more efficient graph reduction strategy relies on a different type of CB-gadget designed specifically for symmetric functions. This gadget is parameterized by positive scalars $a$ and $b$, and includes two auxiliary nodes $e'$ and $e''$. For each node $v \in e$, there is a directed edge from $v$ to $e'$ and a directed edge from $e''$ to $v$, both of weight $a$. Lastly, there is a directed edge from $e'$ to $e''$ of weight $a \cdot b$. This CB-gadget models the following symmetric submodular function:

$$f_{a,b}(A) = a \cdot \min\{|A|, |e\backslash A|, b\}. \tag{24}$$

We previously showed that any submodular cardinality-based function $f_e$ on a ground set $e$ can be modeled with a combination of $\lfloor k/2 \rfloor$ symmetric CB-gadgets where $k = |e|$ [9]. Recall that these authors showed how to use $k - 1$ asymmetric CB-gadgets to model more general submodular cardinality-based functions. Although each symmetric CB-gadget has two auxiliary nodes and asymmetric CB-gadgets has one auxiliary node, note that symmetric CB-gadgets have only one extra edge. Therefore, modeling a symmetric function with $\lfloor k/2 \rfloor$ symmetric CB-gadgets is more efficient (i.e., requires fewer edges overall) than using $k - 1$ asymmetric CB-gadgets. The same type of savings is possible when approximately modeling symmetric submodular functions with symmetric CB-gadgets.

## B.2 Sparse reduction for symmetric concave cardinality functions

Previously we defined a function $f_e$ to be a concave cardinality function if $f_e(A) = g_e(|A|)$ for some concave function $g_e$. If $f_e$ is additionally symmetric, then we will use the fact that $f_e(A) = h_e(\min\{|A|, |e\backslash A|\})$ for some concave *and monotonically increasing* function $h_e$. If $k = |e|$, then $f_e$ has only $r = \lfloor k/2 \rfloor$ different output penalties. The symmetric function in (24), which is modeled by the symmetric CB-gadget, can be defined by $f_{a,b}(A) = h_{a,b}(\min\{|A|, |e\backslash A|\})$ where

$$h_{a,b}(x) = a \cdot \min\{x, b\}. \tag{25}$$

If we combine multiple functions of the form (25) for different parameters $(a, b)$, then we reach the symmetric analog of the combined gadget function from Definition 1.

**Definition 2** *For an integer $k$ and $r = \lfloor k/2 \rfloor$, a type-$r$ symmetric combined gadget function ($r$-Sym-CGF) of order $J$ is a function $\ell: [0, r] \to \mathbb{R}^+$ of the form:*

$$\ell(x) = \sum_{j=1}^{J} a_j \min\{x, b_j\}, \tag{26}$$

*where the parameters $(a_j)$ and $(b_j)$ satisfy*

$$b_j > 0, a_j > 0 \text{ for all } j \in [J] \tag{27}$$
$$b_j < b_{j+1} \text{ for } j \in [J - 1] \tag{28}$$
$$b_J \le r. \tag{29}$$

We similarly obtain have a symmetric analog of the sparse approximate reduction problem.

**Definition 3** *Let $h\colon [0,r] \to \mathbb{R}^+$ be a nonnegative concave increasing function and fix $\varepsilon \geq 0$. The symmetric sparsest approximate reduction (Sym-SpAR) problem seeks a type-$r$ symmetric CGF $\ell$ with minimum order $J$ so that*

$$h(i) \leq \ell(i) \leq (1 + \varepsilon)h(i) \text{ for all } i \in \{0, 1, 2, \dots r\}. \tag{30}$$

### B.3 Connection to piecewise linear approximation

Just as we did for asymmetric functions, we can relate the symmetric sparse approximate reduction (Sym-SpAR) problem to piecewise linear function approximation. The equivalence result and resulting piecewise linear approximation problem is very similar in spirit, though a few subtle and important changes are required due to differences in the symmetric and asymmetric CB-gadgets. For the symmetric case we are approximating a *monotonic* concave function on an interval $[0, \lfloor k/2 \rfloor]$, rather than an arbitrary concave function on an interval $[0, k]$. When approximating such a curve, it ends up being important for the last linear piece to have a slope of zero, otherwise the resulting graph reduction for $f_e$ will have one more CB-gadget that is strictly necessary, and therefore will not quite be optimally sparse. Asymptotically this makes little difference, but in practice, including one more CB-gadget than necessary for each function $f_e$ will lead to an unnecessary increase in runtime. In order to ensure we in fact optimally solve Sym-SpAR and implement the most efficient approach, we carefully outline the necessary changes we need to make for symmetric functions.

The class of piecewise linear functions we consider for the symmetric problem is slightly more specific than the functions we used for the general case, so we include a precise definition.

**Definition 4** *For $r \in \mathbb{N}$, $\mathcal{F}_r$ is the class of functions $f\colon [0, \infty] \longrightarrow \mathbb{R}_+$ such that:*

1. $f(0) = 0$

2. $f$ *is a constant for all $x \geq r$*

3. $f$ *is increasing: $x_1 \leq x_2 \implies f(x_1) \leq f(x_2)$*

4. $f$ *is piecewise linear*

5. $f$ *is concave (and hence, continuous).*

Lemma 1 in the main text provided an exact relationship between asymmetric combined gadget functions (Definition 1) and a certain class of piecewise linear functions. The following results are the analog for symmetric combined gadget functions.

**Lemma 7** *The function $\ell$ in (26) is in the class $\mathcal{F}_r$, and has exactly $J$ positive sloped linear pieces, and one linear piece of slope zero.*

**Proof** Define $b_0 = 0$ for notational convenience. The first three conditions in Definition 4 can be seen by inspection, recalling that $0 < a_j$ and $0 < b_j \leq r$ for all $j \in [J]$. Observe that $\ell$ is linear over the interval $[b_{i-1}, b_i)$ for $i \in [J]$, since for $x \in [b_{i-1}, b_i)$,

$$\ell(x) = \sum_{j=1}^{J} a_j \cdot \min\{x, b_j\} = \sum_{j=1}^{i-1} a_j b_j + x \cdot \sum_{j=i}^{J} a_j.$$

In other words, the $i$th linear piece of $\ell$, defined over $x \in [b_{i-1}, b_i)$ is given by $\ell^{(i)}(x) = I_i + S_i x$, where the intercept and slope terms are given by $I_i = \sum_{j=1}^{i-1} a_j b_j$ and $S_i = \sum_{j=i}^{J} a_j$. For the first $J$ intervals of the form $[b_{i-1}, b_i)$, the slopes are always positive but strictly decreasing. Thus, there are exactly $J$ positive sloped linear pieces. The final linear piece is a flat line, since $\ell(x) = \sum_{j=1}^{J} a_j b_j$ for all $x \geq b_J$. The concavity of $\ell$ follows directly from the fact that it is a continuous and piecewise linear function with decreasing slopes. $\square$

**Lemma 8** *Let $\ell'$ be a function in $\mathcal{F}_r$ with $J + 1$ linear pieces. Let $b_i$ denote the $i$th breakpoint of $\ell'$, and $m_i$ denote the slope of the $i$th linear piece of $\ell$. Define vectors $\boldsymbol{a}, \boldsymbol{b} \in \mathbb{R}^J$ where $\boldsymbol{b}(i) = b_i$ and $\boldsymbol{a}(i) = a_i = m_i - m_{i+1}$ for $i \in [J]$. Then $\ell'$ is the type-$r$ CGF of order $J$ parameterized by vectors $(\boldsymbol{a}, \boldsymbol{b})$.*

**Proof** Since $\ell'$ is in $\mathcal{F}_r$, it has $J$ positive-sloped linear pieces and one flat linear piece, and therefore it has exactly $J$ breakpoints: $0 < b_1 < b_2 < \ldots < b_J$. Let $\mathbf{b} = (b_j)$ be the vector storing these breakpoints. For convenience we define $b_0 = 0$, though $b_0$ is not stored in $\mathbf{b}$. By definition, $\ell'$ is constant for all $x \geq r$, which implies that $b_J \leq r$.

Let $\ell_i = \ell'(b_i)$. For $i \in [J]$, the positive slope of the $i$th linear piece of $\ell'$, which occurs in the range $[b_{i-1}, b_i]$, is given by

$$m_i = \frac{\ell_i - \ell_{i-1}}{b_i - b_{i-1}}. \tag{31}$$

The $i$th linear piece of $\ell'$ is given by

$$\ell^{(i)}(x) = m_i(x - b_{i-1}) + \ell_{i-1} \qquad \text{for } x \in [b_{i-1}, b_i]. \tag{32}$$

The last linear piece of $\ell'$ is a flat line over the interval $x \in [b_J, \infty)$, i.e., $m_{J+1} = 0$. Since $\ell'$ has positive and strictly decreasing slopes, we can see that $a_i = m_i - m_{i+1} > 0$ for all $i \in [J]$.

Let $\hat{\ell}$ be the type-$r$ CGF of order-$J$ constructed from vectors $(\mathbf{a}, \mathbf{b})$:

$$\hat{\ell}(x) = \sum_{j=1}^{J} a_j \cdot \min\{x, b_j\}. \tag{33}$$

We must check that $\hat{\ell} = \ell'$. By Lemma 7, we know that $\hat{\ell}$ is in $\mathcal{F}_r$ and has exactly $J + 1$ linear pieces. The functions will be the same, therefore, if they share the same values at breakpoints. Evaluating $\hat{\ell}$ at an arbitrary breakpoint $b_i$ gives:

$$\hat{\ell}(b_i) = \left(\sum_{j=1}^{i-1} a_j \cdot b_j\right) + b_i \cdot \left(\sum_{j=i}^{J} a_j\right) = \left(\sum_{j=1}^{i-1} a_j \cdot b_j\right) + b_i \cdot m_i. \tag{34}$$

We first confirm that the functions coincide at the first breakpoint:

$$\hat{\ell}(b_1) = b_1 \cdot m_1 = b_1 \cdot \frac{\ell_1 - \ell_0}{b_1 - b_0} = b_1 \frac{\ell_1}{b_1} = \ell_1.$$

For any fixed $i \in \{2, 3, \ldots, J\}$,

$$\begin{aligned}
\hat{\ell}(b_i) - \hat{\ell}(b_{i-1}) &= \left(\sum_{j=1}^{i-1} a_j b_j\right) + b_i m_i - \left(\sum_{j=1}^{i-2} a_j b_j\right) - b_{i-1} m_{i-1} \\
&= a_{i-1} b_{i-1} + b_i m_i - b_{i-1} m_{i-1} \\
&= (m_{i-1} - m_i) b_{i-1} + b_i m_i - b_{i-1} m_{i-1} \\
&= m_i(b_i - b_{i-1}) = \ell_i - \ell_{i-1}.
\end{aligned}$$

Since $\ell'(b_1) = \hat{\ell}(b_1)$ and $\ell'(b_i) - \ell'(b_{i-1}) = \hat{\ell}(b_i) - \hat{\ell}(b_{i-1})$ for $i \in \{2, 3, \ldots, t\}$, we have $\ell'(b_i) = \hat{\ell}(b_i)$ for $i \in [J]$. Therefore, $\ell'$ and $\hat{\ell}$ are the same piecewise linear function. $\qquad \square$

## B.4 Other adjustments for symmetric function graph reductions

Given the equivalence results in Lemma 7 and 8, we can see again that in order to model a symmetric concave cardinality function using symmetric CB-gadgets, we need to solve a piecewise linear approximation problem. Specifically, we need to approximate a concave and increasing function $h$ at integer points $\{0, 1, 2, \ldots, r\}$ with a minimum number of positive sloped linear pieces, and one flat linear piece. To do so, we run Algorithms 1 and 2 to get a collection of lines $\mathcal{L}$, and make the following adjustments to ensure the last linear pieces is flat:

- If the last line in $\mathcal{L}$ has negative slope, remove it.
- If the last two lines provide a $(1 + \varepsilon)$-approximation at the point $\{r - 1, h(r - 1)\}$, and the last line has a positive slope, remove the last line.

Whether or not we apply one of the above two steps, we finish by adding the line $L(x) = h(r)$ to $\mathcal{L}$. This adjustment ensures that we still provide a $(1 + \varepsilon)$-approximation at all integers $\{0, 1, \ldots r\}$, we include one line of slope zero, and we do not keep more *positive sloped* lines than necessary.

We also apply slight adjustments to our graph construction procedure in the case of a symmetric concave cardinality function $f_e$. We apply the same basic approach as in Algorithm 5, except we use the symmetric CB-gadget construction and use Lemma 8 to determine parameters for these CB-gadgets from the piecewise linear approximation. Finally, observe that the asymptotic bounds we prove for approximating certain concave functions with piecewise linear approximations in Section 3.3 also apply to the symmetric case, which only differs in how the last linear piece in the approximation is found. Similarly, the asymptotic number of auxiliary nodes and edges we need to model different kinds of concave cardinality functions does not change for the symmetric case, as the symmetric graph construction and the general asymmetric construction differ only by a factor of two in terms of the number of auxiliary nodes and edges needed.

## C    Experiment Details

We implemented our sparse reduction techniques in the Julia programming language, and used an implementation of the push-relabel maximum flow algorithm to solve the minimum $s$-$t$ cuts for SPARSECARD. To ensure reducibility, we provide additional details for our experimental results.

### C.1    Parameter Settings for Image Segmentation Experiments

The continuous optimization methods for DSFM that we compare against are implemented in C++ with a MATLAB front end. The Incidence Relation AP (IAP) method is an improved version of the AP method [8]. Li and Milenkovic [5] showed that the runtime of the method can be significantly faster if one accounts for so-called *incidence relations*, which describe sets of nodes that define the support of a component function. In our experiments we also ran the standard AP algorithm as implemented by Li and Milenkovic, but this always performed noticeably worse that IAP in practice, so we only report results for IAP. Neither of these methods require setting any hyperparameters.

Li and Milenkovic [5] also showed that accounting for incidence relations leads to improved *parallel* runtimes for ACDM and RCDM, but this does not improve serial runtimes. To simulate improved parallel runtimes, the implementations ACDM and RCDM of these authors include a parallelization parameter $\alpha = K/R$, where $K$ is the number of projections performed in an inner loop of these methods, and $R$ is the number of component functions. In theory, the $K$ projections could be performed in parallel, leading to faster overall runtimes. The comparative parallel performance between methods can be simulated by seeing how quickly the methods converge in terms of the number of total projections performed. Note however that the implementations themselves are serial, and only simulate what could happen in a parallel setting.

In our experiments our goal is to obtain the fastest possible serial runtimes. Li and Milenkovic [5] demonstrated that the minimum number of total projections needed to achieve convergence to within a small tolerance is typically achieved when $\alpha$ is quite small. When projections are performed in parallel, choosing a larger $\alpha$ may still be advantageous. However, since our goal is to obtain the fast serial runtimes, we chose a small value $\alpha = 0.01$, based on the results of Li and Milenkovic. We also tried larger and smaller values of $\alpha$ in post-hoc experiments on all four instances of DSFM, though this led to little variation in performance.

In addition to $\alpha$, ACDM relies on an empirical parameter $c$ controlling the number of iterations in an outer loop. We used the recommended default parameter $c = 10$. In general it is unclear how to set this parameter a priori to obtain better than default behavior on a given DSFM instance. In order to highlight the strength of SPARSECARD relative to ACDM, we additionally tried post-hoc tuning of $c$ on each dataset to see how much this could affect results. We ran ACDM on each of the four instances of DSFM (2 image datasets $\times$ 2 superpixel segmentations each) for all $c \in \{10, 25, 50, 100, 200\}$, three different times, for $50R$ projections (i.e., on average we visit and perform a projection step at each component function 50 times). This took roughly 30-45 seconds for each run. We then computed the average duality gap for each $c$ and each instance over the three trials, and re-ran the algorithm for even longer using the best value of $c$ for each instance. The result is shown as ACDM-best in Figure 1 in the main text. SPARSECARD still maintains a clear advantage over this method on the instances

that involve 200 superpixels (i.e., very large region potentials). Our method also obtains better approximations for the *smallplant* dataset with 500 superpixels for the first 40 seconds, after which point ACDM-best converges to the optimal solution. Nevertheless, we would not have been able to see this improved behavior without post-hoc tuning of the hyperparameter $c$ for ACDM. Meanwhile, SPARSECARD does not rely on any parameter except $\varepsilon$, and it is very easy to understand how setting this parameter affects the algorithm as it directly controls the sparsity and a priori approximation guarantee for our method.

We remark finally that that Li and Milenkovic [5] also considered and implemented an alternate version of RCDM (RCDM-greedy) with a greedy sampling strategy for visiting component functions in the method's inner loop. Despite being advantageous for parallel implementations, we found that in practice that this method had worse serial runtimes, so we did not report results for it.

## C.2    Hypergraph Localized Clustering Experiments

**Background on HyperLocal.** For our local hypergraph clustering experiments, we inserted SPARSECARD as a subroutine into the method HYPERLOCAL, which finds a cluster $S$ in a hypergraph $\mathcal{H} = (V, E)$ that is localized around an input set $Z \subset V$. It does so by minimizing the following ratio cut objective:

$$\phi(S) = \frac{\mathbf{cut}_{\mathcal{H}}(S)}{\mathbf{vol}(Z \cap S) - \beta \mathbf{vol}(\bar{Z} \cap S)}, \text{ subject to } \mathbf{vol}(\bar{Z} \cap S) \geq 0. \tag{35}$$

Here, $\bar{Z} = V \backslash Z$ denotes the complement set of $Z$. For a node set $T \subseteq V$, $\mathbf{vol}(T)$ denotes volume of $T$, i.e., the sum of node degrees. The term $\mathbf{vol}(Z \cap S)$ in the denominator rewards a high overlap between the output cluster $S$ and the input set $Z$. The second term $-\beta \mathbf{vol}(\bar{Z} \cap S)$ is a penalty for including too many nodes outside the input set $Z$. This is tuned by a *locality parameter* $\beta > 0$. For smaller values of $\beta$, the algorithm will explore a larger region in the hypergraph in search for good clusters. The function $\mathbf{cut}_{\mathcal{H}}$ is a generalized hypergraph cut function that can be viewed as a decomposable submodular function

$$\mathbf{cut}_{\mathcal{H}}(S) = \sum_{e \in E} f_e(e \cap S), \tag{36}$$

where $f_e$ determines the penalty for how the hyperedge $e$ is split among two clusters.

HYPERLOCAL minimizes (35) by solving a sequence of hypergraph $s$-$t$ cut problems, which can also be viewed as DSFM problems. These $s$-$t$ cut problems are solved using the previous exact graph reduction techniques for concave cardinality functions $f_e$ that the authors designed in earlier work [9]. However, the existing implementation of HYPERLOCAL only uses the $\delta$-linear component function, which takes the form

$$f_e(A) = \min\{|A|, |e \backslash A|, \delta\} \text{ for } A \subseteq e, \tag{37}$$

for a parameter $\delta \geq 1$. One of the major benefits of this hyperedge cut penalty is that it can be modeled exactly and sparsely using a single CB-gadget with parameters $a = 1$ and $b = \delta$.

**HyperLocal + SparseCard.** Alternative hyperedge cut penalties have the potential to produce improved results in some applications. However, exact reduction techniques for alternate penalties can become infeasible if they require too many CB-gadgets to model, as this results in a very dense reduced graph. This will no longer be a problem when we use SPARSECARD as a subroutine for HYPERLOCAL. Our updated version of HYPERLOCAL takes in a parameter $\varepsilon \geq 0$ and any set of concave cardinality penalties $\{f_e\}_{e \in E}$. We then use SPARSECARD (i.e., a sparse approximate reduction followed by solving a graph $s$-$t$ cut problem), to solve the sequence of hypergraph $s$-$t$ cut problems needed to minimize a ratio objective of the form (35). We specifically consider the following three alternative cut penalties in our experiments:

$$\text{Clique penalty: } f_e(A) = (|e| - 1)^{-1}|A||e \backslash A|$$

$$\text{Square root penalty: } f_e(A) = \sqrt{\min\{|A|, |e \backslash A|\}}$$

$$\text{Sublinear power } (x^{0.9}) \text{ penalty: } f_e(A) = (\min\{|A|, |e \backslash A|\})^{0.9}.$$

All of these penalties require $O(|e|)$ CB-gadgets to model exactly using previous reduction techniques. These penalties satisfy the normalizing condition that $f_e(A) = 1$ when $|A| = 1$, though if desired

they could be scaled by a constant. Scaling the clique penalty $|A||e \backslash A|$ by $(|e| - 1)^{-1}$ has several other additional desirable properties that have led to its use in numerous other hypergraph clustering frameworks [11, 3, 1].

**Experimental setup and parameter settings.** We follow the same experimental setup as our previous work [10] for detecting localized clusters in the Stackoverflow hypergraph. We focus on the same set of 45 local clusters, all of which are question topics involving between 2,000 and 10,000 nodes. For each cluster, we generate a random seed set by selecting 5% of the nodes in the target cluster uniformly at random, and then add neighboring nodes to the seed set to grow it into a larger input set $Z$ to use for HYPERLOCAL (see [10] for details). We set $\delta = 5000$ for the $\delta$-linear hyperedge cut function and set the locality parameters to be $\beta = 1.0$ for all experiments. With this setup, using HYPERLOCAL with the $\delta$-linear penalty will then reproduce our original experimental setup [10]. Our goal is to show how using SPARSECARD leads to fast and often improved results for alternative penalties that could not previously been used.

**Summary of experimental findings.** In the main text we showed average runtimes and F1 scores for cluster detection across the 45 clusters using four hyperedge cut penalties. Using $\delta$-linear penalty corresponds to a previous approach. Using the clique, square root, and sublinear power penalties demonstrate the utility of SPARSECARD. We highlight three main takeaways from these results in Table 3 in the main text.

1. SPARSECARD *leads to improved results that would not be possible with previous techniques.* In particular, the clique and sublinear power penalty often obtain better F1 scores than the $\delta$-linear penalty, but we would not be able to use these without our sparse reduction techniques. Given that the hypergraph has thousands of hyperedges with over a thousand nodes, using $O(|e|^2)$ edges to model these splitting penalties for each $e \in E$ is infeasible. Furthermore, we previously demonstrated that discarding all hyperedges above 50 nodes and performing an exact clique expansion leads to poor detection results (and even so the reduced graph is quite dense) [10].

2. *Approximate reductions lead to significantly improved runtimes, while still approximating the original hyperedge cut function extremely well.* We saw almost no difference in F1 scores when using $\varepsilon = 1.0$, $\varepsilon = 0.1$, and $\varepsilon = 0.01$, although larger $\varepsilon$ values led to much faster runtimes. This matches our observation in image segmentation experiments that for a parameter $\varepsilon$, SPARSECARD will tend to provide much better than just a $(1 + \varepsilon)$ approximation in practice.

3. *Our approximate reductions are efficient and useful even for the hardest functions to model.* Theorem 4 in the main text showed that the square root function exhibits worst case behavior in terms of the number of CB-gadgets needed to approximately model it. Nevertheless, our results on the Stackoverflow hypergraph indicate that we can obtain approximations and runtimes for this penalty that are nearly as fast as other penalties. For the Stackoverflow dataset, this penalty does not perform particularly well in terms of F1 scores, but this shows us SPARSECARD can provide an efficient and useful way to model any concave cardinality penalty that one may encounter in different applications.

**Extended experimental results.** In order to further confirm the three main findings highlighted above, we provide extended results on the Stackoverflow hypergraph. Results in the main text are shown for two values of $\varepsilon$, a single seed set for each cluster, and were obtained by running experiments on a laptop with 8GB of RAM. In order to run a larger number of experiments, we additionally ran experiments on a machine with 4 x 18-core, 3.10 GHz Intel Xeon gold processors with 1.5 TB RAM. We generated 10 different randomly selected seed sets for each cluster, and ran each method for each of the ten seed sets on each cluster. We used three different approximation parameters for the three alternative penalties, $\varepsilon \in \{1.0, 0.1, 0.01\}$. This amounts to solving 450 localized clustering experiments using ten different methods.

In Figure 2 we show the change in runtime and the change in F1 score that results from using different values of $\varepsilon$ for each alternative hyperedge cut function. Using a larger values $\varepsilon = 1.0$ leads to significantly faster runtimes, but virtually indistinguishable F1 scores. In Figure 1 we show the mean F1 score for each cluster (across the ten random seed sets) obtained by the $\delta$-linear penalty and the three alternative penalties when $\varepsilon = 1.0$. The clique penalty obtained the highest average F1 score on 26 clusters, the $\delta$-linear obtained the highest average score on 10 clusters, and the sublinear penalty

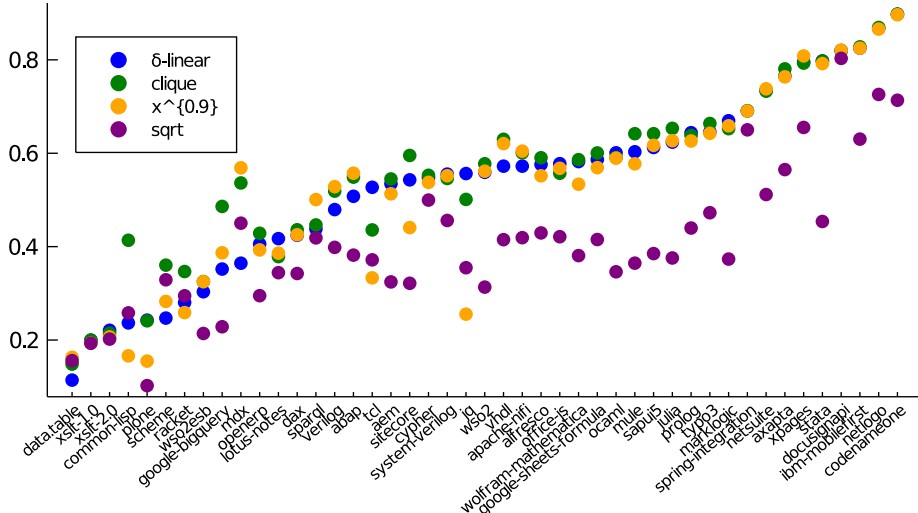

Figure 1: Average F1 score obtained for each localized clustering using 4 different hyperedge cut penalties. For clique, $x^{0.9}$, and the square root penalties, we used an approximate reduction with $\varepsilon = 1.0$. The clique penalty had the highest average F1 score on 26 clusters, the $\delta$-linear had the highest on 10 clusters, and $x^{0.9}$ had the highest average score on the remaining 10 clusters.

obtained the best average score on the remaining 9 clusters. In Tables 1 through 4 we give more detailed results for each individual cluster.

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

Table 1: Runtime and F1 scores for detecting local clusters from size 2018 to size 2536 in a Stackoverflow question hypergraph using HyperLocal [9] + SPARSECARD. For each cluster, we generated 10 random seed sets by randomly sampling 5% of the cluster, and ran HyperLocal + SPARSECARD with five different hyperedge cut penalty functions on each seed set. We report standard deviation across the ten seed sets for each cluster. Without SPARSECARD, it would not be possible to run the experiment using the *clique*, $x^{0.9}$, or *sqrt* cut penalties, as existing exact graph reduction strategies produce a graph that is far too dense. The *# Best* column indicates the number of times, out of the ten seed sets, that using the given penalty function leads to the highest F1 score for each cluster. We highlight the best results in bold. In some cases, answers are the same to within the reported 3 significant figures, but the bolded number highlights which method had slightly better unrounded F1 scores.

| Cluster | Size | Penalty | F1 | | Runtime | | # Best |
|---|---|---|---|---|---|---|---|
| system-verilog | 2018 | $\delta$-linear | **0.555** | $\pm 0.02$ | **5.9** | $\pm 3.8$ | **7** |
| | | clique | 0.546 | $\pm 0.02$ | 8.0 | $\pm 2.0$ | 0 |
| | | sqrt | 0.456 | $\pm 0.05$ | 7.4 | $\pm 2.2$ | 0 |
| | | $x^{0.9}$ | 0.552 | $\pm 0.01$ | 6.2 | $\pm 0.8$ | 3 |
| abap | 2056 | $\delta$-linear | 0.508 | $\pm 0.12$ | 11.8 | $\pm 9.0$ | 3 |
| | | clique | 0.549 | $\pm 0.06$ | 12.9 | $\pm 7.0$ | 0 |
| | | sqrt | 0.382 | $\pm 0.07$ | 20.4 | $\pm 12.5$ | 0 |
| | | $x^{0.9}$ | **0.557** | $\pm 0.06$ | **9.8** | $\pm 5.4$ | **7** |
| axapta | 2074 | $\delta$-linear | 0.766 | $\pm 0.05$ | **22.4** | $\pm 13.3$ | 0 |
| | | clique | **0.781** | $\pm 0.04$ | 23.6 | $\pm 15.2$ | **9** |
| | | sqrt | 0.565 | $\pm 0.09$ | 41.6 | $\pm 31.2$ | 0 |
| | | $x^{0.9}$ | 0.764 | $\pm 0.05$ | 26.1 | $\pm 16.8$ | 1 |
| apache-nifi | 2092 | $\delta$-linear | 0.572 | $\pm 0.06$ | **6.7** | $\pm 2.1$ | 1 |
| | | clique | 0.601 | $\pm 0.07$ | 7.6 | $\pm 1.6$ | 3 |
| | | sqrt | 0.419 | $\pm 0.13$ | 8.4 | $\pm 1.6$ | 0 |
| | | $x^{0.9}$ | **0.605** | $\pm 0.07$ | 6.9 | $\pm 1.2$ | **6** |
| google-sheets-formula | 2142 | $\delta$-linear | 0.587 | $\pm 0.19$ | **4.9** | $\pm 1.0$ | 1 |
| | | clique | **0.601** | $\pm 0.19$ | 8.7 | $\pm 2.0$ | **7** |
| | | sqrt | 0.415 | $\pm 0.13$ | 8.2 | $\pm 2.6$ | 0 |
| | | $x^{0.9}$ | 0.569 | $\pm 0.17$ | 5.7 | $\pm 1.0$ | 2 |
| office-js | 2402 | $\delta$-linear | **0.578** | $\pm 0.04$ | **7.2** | $\pm 2.0$ | **9** |
| | | clique | 0.557 | $\pm 0.05$ | 7.7 | $\pm 1.2$ | 0 |
| | | sqrt | 0.421 | $\pm 0.05$ | 9.1 | $\pm 2.2$ | 0 |
| | | $x^{0.9}$ | 0.568 | $\pm 0.04$ | 7.8 | $\pm 1.1$ | 1 |
| netlogo | 2520 | $\delta$-linear | 0.868 | $\pm 0.01$ | **20.8** | $\pm 12.1$ | 2 |
| | | clique | **0.869** | $\pm 0.01$ | 22.0 | $\pm 15.2$ | **6** |
| | | sqrt | 0.726 | $\pm 0.15$ | 34.3 | $\pm 20.5$ | 0 |
| | | $x^{0.9}$ | 0.866 | $\pm 0.02$ | 23.0 | $\pm 14.3$ | 2 |
| dax | 2528 | $\delta$-linear | 0.424 | $\pm 0.03$ | **7.3** | $\pm 2.3$ | 0 |
| | | clique | **0.436** | $\pm 0.04$ | 8.3 | $\pm 1.0$ | **10** |
| | | sqrt | 0.342 | $\pm 0.05$ | 9.1 | $\pm 2.4$ | 0 |
| | | $x^{0.9}$ | 0.425 | $\pm 0.04$ | 8.0 | $\pm 1.1$ | 0 |
| plone | 2536 | $\delta$-linear | **0.243** | $\pm 0.14$ | **2.8** | $\pm 0.3$ | **7** |
| | | clique | 0.241 | $\pm 0.14$ | 5.2 | $\pm 0.9$ | 3 |
| | | sqrt | 0.102 | $\pm 0.04$ | 8.0 | $\pm 2.0$ | 0 |
| | | $x^{0.9}$ | 0.155 | $\pm 0.1$ | 4.2 | $\pm 0.5$ | 0 |

Table 2: Results for Stackoverflow clusters from size 2574 to size 3506.

| Cluster | Size | Penalty | F1 | | Runtime | | # Best |
|---|---|---|---|---|---|---|---|
| netsuite | 2574 | $\delta$-linear | 0.735 | ±0.06 | 28.8 | ±18.7 | 4 |
| | | clique | 0.733 | ±0.07 | 28.5 | ±17.1 | 1 |
| | | sqrt | 0.512 | ±0.12 | 37.5 | ±24.1 | 0 |
| | | $x^{0.9}$ | **0.738** | ±0.07 | **28.1** | ±18.1 | **5** |
| jq | 2596 | $\delta$-linear | **0.557** | ±0.14 | **3.8** | ±1.0 | **10** |
| | | clique | 0.501 | ±0.14 | 5.7 | ±0.9 | 0 |
| | | sqrt | 0.355 | ±0.09 | 8.0 | ±1.7 | 0 |
| | | $x^{0.9}$ | 0.255 | ±0.13 | 4.7 | ±0.7 | 0 |
| marklogic | 2612 | $\delta$-linear | **0.67** | ±0.14 | **4.7** | ±1.2 | **7** |
| | | clique | 0.653 | ±0.15 | 7.1 | ±1.6 | 2 |
| | | sqrt | 0.373 | ±0.12 | 8.7 | ±1.8 | 0 |
| | | $x^{0.9}$ | 0.659 | ±0.14 | 5.7 | ±0.4 | 1 |
| alfresco | 2694 | $\delta$-linear | 0.576 | ±0.13 | **13.0** | ±7.6 | 1 |
| | | clique | **0.59** | ±0.11 | 15.5 | ±7.9 | **9** |
| | | sqrt | 0.429 | ±0.09 | 20.4 | ±9.3 | 0 |
| | | $x^{0.9}$ | 0.552 | ±0.18 | 15.0 | ±8.5 | 0 |
| lotus-notes | 2877 | $\delta$-linear | **0.417** | ±0.06 | **4.4** | ±1.3 | **9** |
| | | clique | 0.379 | ±0.06 | 5.5 | ±0.9 | 0 |
| | | sqrt | 0.344 | ±0.06 | 8.5 | ±1.5 | 0 |
| | | $x^{0.9}$ | 0.386 | ±0.06 | 5.1 | ±0.6 | 1 |
| stata | 2907 | $\delta$-linear | 0.798 | ±0.05 | **6.6** | ±1.4 | 1 |
| | | clique | **0.798** | ±0.05 | 10.0 | ±2.0 | **5** |
| | | sqrt | 0.454 | ±0.05 | 9.2 | ±2.4 | 0 |
| | | $x^{0.9}$ | 0.792 | ±0.05 | 8.6 | ±1.6 | 4 |
| wso2esb | 2912 | $\delta$-linear | 0.303 | ±0.05 | 7.8 | ±6.6 | 2 |
| | | clique | **0.325** | ±0.09 | 8.7 | ±5.0 | **4** |
| | | sqrt | 0.214 | ±0.08 | 11.6 | ±5.9 | 0 |
| | | $x^{0.9}$ | 0.325 | ±0.1 | **7.3** | ±2.7 | 4 |
| mdx | 3007 | $\delta$-linear | 0.365 | ±0.12 | 8.4 | ±4.2 | 0 |
| | | clique | 0.536 | ±0.04 | 8.9 | ±3.3 | 2 |
| | | sqrt | 0.45 | ±0.04 | 9.5 | ±3.4 | 0 |
| | | $x^{0.9}$ | **0.569** | ±0.03 | **7.5** | ±1.8 | **8** |
| docusignapi | 3348 | $\delta$-linear | 0.82 | ±0.01 | **37.8** | ±12.9 | 1 |
| | | clique | 0.82 | ±0.01 | 41.3 | ±16.5 | 0 |
| | | sqrt | 0.803 | ±0.06 | 53.7 | ±21.8 | **6** |
| | | $x^{0.9}$ | **0.821** | ±0.0 | 41.7 | ±12.8 | 3 |
| xslt-2.0 | 3426 | $\delta$-linear | **0.221** | ±0.08 | 4.8 | ±1.4 | **7** |
| | | clique | 0.215 | ±0.08 | 6.7 | ±1.3 | 0 |
| | | sqrt | 0.202 | ±0.06 | 8.2 | ±1.5 | 0 |
| | | $x^{0.9}$ | 0.207 | ±0.06 | **4.6** | ±0.6 | 3 |
| wolfram-mathematica | 3478 | $\delta$-linear | 0.582 | ±0.04 | **4.5** | ±1.0 | 2 |
| | | clique | **0.586** | ±0.05 | 6.9 | ±0.7 | 3 |
| | | sqrt | 0.381 | ±0.04 | 9.2 | ±2.1 | 0 |
| | | $x^{0.9}$ | 0.534 | ±0.13 | 5.3 | ±0.8 | **5** |
| aem | 3506 | $\delta$-linear | 0.535 | ±0.07 | 25.4 | ±27.1 | 1 |
| | | clique | **0.545** | ±0.1 | 24.4 | ±21.8 | **7** |
| | | sqrt | 0.324 | ±0.14 | 31.8 | ±23.1 | 0 |
| | | $x^{0.9}$ | 0.513 | ±0.12 | **19.9** | ±17.5 | 2 |

Table 3: Results for Stackoverflow clusters from size 3620 to size 5476.

| Cluster | Size | Penalty | F1 | Runtime | # Best |
|---|---|---|---|---|---|
| sparql | 3620 | $\delta$-linear | 0.438 ±0.03 | **8.6** ±4.2 | 0 |
| | | clique | 0.446 ±0.09 | 9.9 ±1.8 | 3 |
| | | sqrt | 0.419 ±0.04 | 10.5 ±3.6 | 1 |
| | | $x^{0.9}$ | **0.501** ±0.06 | 8.7 ±2.3 | **6** |
| codenameone | 3677 | $\delta$-linear | 0.898 ±0.02 | **15.6** ±9.7 | **4** |
| | | clique | **0.898** ±0.02 | 20.4 ±12.2 | 4 |
| | | sqrt | 0.713 ±0.11 | 24.7 ±19.9 | 0 |
| | | $x^{0.9}$ | 0.897 ±0.02 | 18.8 ±10.9 | 2 |
| vhdl | 4135 | $\delta$-linear | 0.572 ±0.05 | **8.2** ±6.4 | 1 |
| | | clique | **0.63** ±0.03 | 10.9 ±6.8 | **5** |
| | | sqrt | 0.415 ±0.03 | 13.8 ±5.9 | 0 |
| | | $x^{0.9}$ | 0.621 ±0.04 | 8.5 ±5.1 | 4 |
| verilog | 4153 | $\delta$-linear | 0.479 ±0.02 | **6.4** ±1.7 | 1 |
| | | clique | 0.519 ±0.04 | 8.2 ±1.2 | 1 |
| | | sqrt | 0.398 ±0.07 | 9.6 ±2.2 | 0 |
| | | $x^{0.9}$ | **0.528** ±0.05 | 7.1 ±1.0 | **8** |
| racket | 4188 | $\delta$-linear | 0.28 ±0.11 | **4.0** ±0.7 | 1 |
| | | clique | **0.347** ±0.14 | 6.0 ±1.0 | **7** |
| | | sqrt | 0.295 ±0.13 | 8.6 ±1.9 | 1 |
| | | $x^{0.9}$ | 0.259 ±0.14 | 4.7 ±0.6 | 1 |
| xslt-1.0 | 4480 | $\delta$-linear | 0.2 ±0.05 | 5.5 ±1.4 | **5** |
| | | clique | **0.2** ±0.05 | 6.9 ±1.2 | 2 |
| | | sqrt | 0.193 ±0.04 | 8.7 ±2.5 | 1 |
| | | $x^{0.9}$ | 0.193 ±0.04 | **4.7** ±0.8 | 2 |
| common-lisp | 4632 | $\delta$-linear | 0.237 ±0.11 | **4.3** ±0.8 | 0 |
| | | clique | **0.414** ±0.09 | 5.9 ±0.7 | **10** |
| | | sqrt | 0.258 ±0.07 | 8.5 ±2.4 | 0 |
| | | $x^{0.9}$ | 0.166 ±0.11 | 5.4 ±0.6 | 0 |
| sapui5 | 4746 | $\delta$-linear | 0.612 ±0.09 | 23.8 ±23.3 | 2 |
| | | clique | **0.642** ±0.06 | 27.6 ±25.0 | **7** |
| | | sqrt | 0.385 ±0.12 | 39.7 ±29.0 | 0 |
| | | $x^{0.9}$ | 0.617 ±0.08 | **22.8** ±19.9 | 1 |
| xpages | 4818 | $\delta$-linear | 0.796 ±0.05 | **35.5** ±19.0 | 2 |
| | | clique | 0.793 ±0.06 | 36.0 ±19.4 | 2 |
| | | sqrt | 0.655 ±0.13 | 53.1 ±29.9 | 1 |
| | | $x^{0.9}$ | **0.808** ±0.06 | 37.4 ±21.2 | 5 |
| openerp | 4884 | $\delta$-linear | 0.406 ±0.1 | 8.4 ±5.3 | 2 |
| | | clique | **0.429** ±0.14 | 9.6 ±4.1 | **6** |
| | | sqrt | 0.295 ±0.09 | 15.5 ±5.9 | 0 |
| | | $x^{0.9}$ | 0.393 ±0.16 | **8.2** ±3.0 | 2 |
| julia | 5295 | $\delta$-linear | 0.624 ±0.08 | **8.9** ±2.7 | 3 |
| | | clique | **0.653** ±0.05 | 11.3 ±3.0 | 3 |
| | | sqrt | 0.376 ±0.05 | 19.1 ±4.2 | 0 |
| | | $x^{0.9}$ | 0.627 ±0.07 | 9.4 ±1.4 | **4** |
| sitecore | 5476 | $\delta$-linear | 0.543 ±0.18 | 19.9 ±19.1 | 1 |
| | | clique | **0.595** ±0.13 | 17.4 ±13.2 | **9** |
| | | sqrt | 0.322 ±0.13 | 30.3 ±21.3 | 0 |
| | | $x^{0.9}$ | 0.441 ±0.24 | **14.3** ±12.7 | 0 |

Table 4: Results for Stackoverflow clusters from size 5536 to size 9859.

| Cluster | Size | Penalty | F1 | | Runtime | | # Best |
|---|---|---|---|---|---|---|---|
| ibm-mobilefirst | 5536 | $\delta$-linear | 0.825 | $\pm$0.02 | **52.1** | $\pm$37.6 | 2 |
| | | clique | **0.828** | $\pm$0.02 | 58.5 | $\pm$42.0 | **4** |
| | | sqrt | 0.63 | $\pm$0.09 | 80.9 | $\pm$48.0 | 0 |
| | | $x^{0.9}$ | 0.825 | $\pm$0.03 | 57.3 | $\pm$38.4 | 4 |
| ocaml | 5590 | $\delta$-linear | **0.601** | $\pm$0.02 | **6.6** | $\pm$1.4 | **4** |
| | | clique | 0.591 | $\pm$0.04 | 9.1 | $\pm$1.9 | 3 |
| | | sqrt | 0.346 | $\pm$0.04 | 9.4 | $\pm$2.7 | 0 |
| | | $x^{0.9}$ | 0.59 | $\pm$0.04 | 7.0 | $\pm$0.8 | 3 |
| spring-integration | 5635 | $\delta$-linear | 0.691 | $\pm$0.0 | 7.7 | $\pm$2.4 | 0 |
| | | clique | **0.691** | $\pm$0.0 | 12.0 | $\pm$3.3 | **8** |
| | | sqrt | 0.65 | $\pm$0.06 | 17.9 | $\pm$5.4 | 1 |
| | | $x^{0.9}$ | 0.69 | $\pm$0.0 | **7.4** | $\pm$1.3 | 1 |
| tcl | 5752 | $\delta$-linear | **0.527** | $\pm$0.12 | **5.2** | $\pm$0.9 | **8** |
| | | clique | 0.436 | $\pm$0.15 | 7.6 | $\pm$1.5 | 2 |
| | | sqrt | 0.372 | $\pm$0.06 | 11.4 | $\pm$3.1 | 0 |
| | | $x^{0.9}$ | 0.333 | $\pm$0.2 | 6.3 | $\pm$1.1 | 0 |
| mule | 5940 | $\delta$-linear | 0.603 | $\pm$0.15 | 11.8 | $\pm$11.3 | 1 |
| | | clique | **0.642** | $\pm$0.12 | 13.6 | $\pm$9.1 | **9** |
| | | sqrt | 0.365 | $\pm$0.05 | 19.5 | $\pm$9.7 | 0 |
| | | $x^{0.9}$ | 0.577 | $\pm$0.16 | **11.8** | $\pm$8.7 | 0 |
| scheme | 6411 | $\delta$-linear | 0.247 | $\pm$0.1 | 6.0 | $\pm$1.4 | 2 |
| | | clique | **0.36** | $\pm$0.07 | 8.4 | $\pm$2.3 | **5** |
| | | sqrt | 0.329 | $\pm$0.04 | 9.6 | $\pm$1.8 | 2 |
| | | $x^{0.9}$ | 0.283 | $\pm$0.11 | **5.4** | $\pm$1.4 | 1 |
| typo3 | 6414 | $\delta$-linear | 0.646 | $\pm$0.09 | 49.3 | $\pm$37.6 | 1 |
| | | clique | **0.664** | $\pm$0.07 | 41.7 | $\pm$29.1 | **7** |
| | | sqrt | 0.473 | $\pm$0.1 | 52.9 | $\pm$28.2 | 0 |
| | | $x^{0.9}$ | 0.643 | $\pm$0.08 | **37.5** | $\pm$24.3 | 2 |
| cypher | 6735 | $\delta$-linear | 0.547 | $\pm$0.02 | 14.3 | $\pm$2.8 | 2 |
| | | clique | **0.553** | $\pm$0.01 | 13.9 | $\pm$1.9 | **7** |
| | | sqrt | 0.5 | $\pm$0.03 | 20.7 | $\pm$5.7 | 0 |
| | | $x^{0.9}$ | 0.538 | $\pm$0.04 | **12.6** | $\pm$1.7 | 1 |
| wso2 | 7760 | $\delta$-linear | 0.559 | $\pm$0.07 | 17.7 | $\pm$12.0 | 2 |
| | | clique | **0.577** | $\pm$0.05 | 18.6 | $\pm$13.7 | **6** |
| | | sqrt | 0.313 | $\pm$0.07 | 30.4 | $\pm$12.9 | 0 |
| | | $x^{0.9}$ | 0.561 | $\pm$0.07 | **17.0** | $\pm$13.0 | 2 |
| data.table | 8108 | $\delta$-linear | 0.114 | $\pm$0.01 | 5.0 | $\pm$0.9 | 0 |
| | | clique | 0.148 | $\pm$0.04 | 5.9 | $\pm$1.3 | **7** |
| | | sqrt | 0.156 | $\pm$0.02 | 16.2 | $\pm$4.2 | 0 |
| | | $x^{0.9}$ | **0.163** | $\pm$0.02 | **4.7** | $\pm$0.2 | 3 |
| prolog | 9086 | $\delta$-linear | **0.644** | $\pm$0.02 | 11.7 | $\pm$3.7 | **7** |
| | | clique | 0.638 | $\pm$0.03 | 11.6 | $\pm$1.5 | 1 |
| | | sqrt | 0.44 | $\pm$0.03 | 17.1 | $\pm$3.8 | 0 |
| | | $x^{0.9}$ | 0.626 | $\pm$0.04 | **8.8** | $\pm$1.1 | 2 |
| google-bigquery | 9859 | $\delta$-linear | 0.352 | $\pm$0.19 | **7.7** | $\pm$2.8 | 1 |
| | | clique | **0.486** | $\pm$0.11 | 10.0 | $\pm$2.9 | **8** |
| | | sqrt | 0.228 | $\pm$0.07 | 18.4 | $\pm$3.6 | 0 |
| | | $x^{0.9}$ | 0.387 | $\pm$0.18 | 9.4 | $\pm$1.9 | 1 |