# OpenReview forum: "Approximate Decomposable Submodular Function Minimization for Cardinality-Based Components"
_NeurIPS.cc/2021/Conference — NeurIPS 2021 Poster_

### Official Review · Reviewer_uhsD · 2021-07-01

**Rating:** 6
**Confidence:** 4

**Summary:**

This work studies minimizing decomposable submodular functions, which are of the form f(S) = \sum_{e \in E}f_e(S \cap e), where each f_e is a ``simple'' submodular function. In many practical applications, f_e(A) is cardinality based, that is, f_e(A) = g_e(|A|) for some concave function g_e. Several discrete/continuous algorithms have been proposed to minimize such decomposable submodular functions. One combinatorial approach to tackle this problem is to introduce a gadget graph G_e so that, for any A, g_e(|A|) is equal to the minimum cut of G_e subject to A (in addition to some special vertex) being contained in one side of the cut. It is known that any cardinality-based submodular function can be represented this way as a gadget graph of O(|e|) vertices and O(|e|^2) edges.

The main contribution of this work is providing a method for constructing a gadget graph that ``approximates'' cardinality-based submodular function instead of exactly simulating it. Specifically, it is shown that, to obtain a multiplicative error of 1 \pm epsilon, we only need O(epsilon^{-1} |e| log |e|) edges. A lower bound of O(epsilon^{-1/2} |e| log |e|) is also shown. When g(x) = x(k-x), which often appears in practice, it is shown that the number of edges can be further improved to O(\epsilon^{-1/2}log log epsilon^{-1} \cdot |e|). These results are obtained by approximating the concave function g_e with a piecewise linear function l and then representing l as a gadget graph.

Empirical results on image segmentation tasks show that the proposed approximate reduction outputs a much smaller gadget graph compared to the exact reduction without much sacrificing the quality of the solution. For example, for some choice of the parameters, the size of the output graph becomes 0.013 (1/0.013?) times smaller while the multiplicative error is merely 1 \pm 4*10^-3. It is then confirmed that the trade-off of the approximation ratio and runtime of the proposed method is much better than those of other methods. It is also empirically investigated that the quality of hypergraph clustering does not deteriorate much by replacing the objective submodular functions with gadget graphs obtained by the proposed approximate reduction.

**Main Review:**

- The biggest issue in evaluating this work is that I don't fully understand the relevance of this study. I understand that the proposed method outperforms other methods when there are large e's in the input decomposable submodular function. But in practice are there large e's? I understand there are a few of them as in the hypergraph used in the experiment on local clustering, but can't we just ignore those large hyperedges and quickly get a good approximation?
- Related to the question above, in the image segmentation task, what is the max/average size of e?
- It is nice that a detailed comparison of time complexity is provided because it is often implicit in the literature, which blurs the difference between existing methods.


**Time Spent Reviewing:**

3

---

> ### Author Response · Authors · 2021-08-10
> **Reviewer response**
>
> Thank you for the detailed review and the comments! We address your two questions in turn:
>
> "But in practice are there large e's? I understand there are a few of them as in the hypergraph used in the experiment on local clustering, but can't we just ignore those large hyperedges and quickly get a good approximation?"
>
> The idea of just ignoring large hyperedges is indeed a natural thing to try. However, in practice this actually performs quite poorly, so it indeed is important to keep large hyperedges. It is easily missed, but we included the following sentence on line 332 of our paper: "using a clique expansion after simply removing large hyperedges was shown to perform poorly [42]". You can find detailed results in previous work of Veldt et al. [42], that show that performing a clique expansion after removing large hyperedges does poorly. In contrast, we show that the (approximate) clique splitting function does the best overall when we do not discard large hyperedges and use our sparse reduction techniques.
>
> "Related to the question above, in the image segmentation task, what is the max/average size of e?"
>
> Again, the max and average size of e can be quite large, specifically for the superpixel region potentials. When there are 500 superpixels, the max and mean region potential sizes are around 1000 and 500. When there are 200 superpixels, the max and mean increase to roughly 2000 and 1000.
>
> In general, there are many situations where the maximum sized e can be quite large. For more examples, we include a list of references on line 273.

---

### Official Review · Reviewer_tCoX · 2021-07-02

**Rating:** 7
**Confidence:** 4

**Summary:**

This paper presents an algorithm for minimizing submodular functions that are the sum of cardinality-based components. This is a special case of decomposable submodular function minimization. Formally, the problem considered is to minimize $f(S) = \sum\limits_{e\in E} g_e(|S \cap e|)$, where $E\subseteq 2^V$ and $g_e$ are 1-dimensional concave functions. One instance of such functions are the region potentials used in image segmentation, where $g_e(x) = c_e \cdot x (|e| - x)$ for some constant $c_e \geq 0$. Such minimization problems are also considered in hypergraph clustering applications.

The authors propose an approximation algorithm that works by approximately reducing their problem to a graph min cut problem. The fact that concave functions can be _exactly_ represented by graphs is known, and the resulting number of edges in the graph is $O\left(\sum\limits_{e\in E} |e|^2\right)$. The main result of the paper is to show how to reduce the number of edges by allowing _approximate_ reductions, leading to a graph with $\widetilde{O}\left(\frac{1}{\epsilon} \sum\limits_{e\in E} |e|\right)$ edges while incurring an $(1+\epsilon)$-multiplicative error in the objective.

The idea is to simplify each concave function before converting it to a graph. Viewing each concave function as a piecewise linear function, the size of the resulting graph depends on the number of pieces. The authors present an algorithm that is able to find the _best_ (in terms of the number of pieces) $(1+\epsilon)$-approximation to this function. By known results, the optimal number of pieces is $\widetilde{O}\left(\frac{1}{\epsilon}\right)$, thus follows the $\widetilde{O}\left(\frac{1}{\epsilon} \sum\limits_{e\in E} |e|\right)$ bound on the number of edges.

For the special case of the region potentials $g_e(x) = c_e \cdot x (|e| - x)$, the authors are able to get a better bound of $\widetilde{O}\left(\frac{1}{\sqrt{\epsilon}} \sum\limits_{e\in E} |e|\right)$ on the number of edges.

The theoretical results are accompanied by experimental results in the problems of image segmentation and hypergraph clustering. For image segmentation, the results show a speedup of at least 3x in 3 out of 4 experiments compared to previous methods, while it is slower in 1 of the 4 experiments. For hypergraph clustering, the F1 score is improved by 3 percentage points at the expense of a 30% runtime increase.

**Limitations And Societal Impact:**

The limitation that I could find in the paper is that there are not enough experiments to make a conclusive practical evaluation (e.g. only 4 image segmentation examples). The authors have addressed the societal impact.

**Main Review:**

**Originality**

The idea of finding the _optimal_ approximation to the concave functions and then turning the new function into a graph is very nice and original to the best of my knowledge. Also, the experimental work gives original insights into the performance of these algorithms.

**Quality**

I checked most of the theoretical results and they are sound. The experimental section is explained well and seems like a fair analysis between different methods, albeit quite limited (e.g. only 4 image segmentation examples). However, I have doubts regarding the theoretical comparison between methods in Section 4.

-The discussion in lines 214-220 seems misguided to me. The fact that exact algorithms don't give a speedup when the error tolerance increases does _not_ make them worse than approximation algorithms. Additionally, the statement about multiplicative approximation doesn't make much sense to me, in the context of comparing to exact algorithms. I would just remove this paragraph.

-In Section 4, it should be emphasized that all previous methods are more general, as they work with arbitrary submodular functions. This is written in other parts of the paper, but should be emphasized in this section which compares different results. In particular, certain runtimes from previous work can be improved in the case of cardinality functions. As an example, in the $\sum\limits_{e} |e|^2 \theta_e$ dependence in [4], the $\theta_e$ comes from solving problems of the form $\min_S f_e(S) + w(S)$, where $w$ is linear. If $f_e$ is a cardinality function, these can be easily solved by sorting, immediately improving this term to $\sum\limits_{e} |e|^2$.

-On the same note, it seems to me that it might be possible to plug the graph approximation results in the current paper into [4] in an almost black-box way to replace the existing graph approximation routine, in order to give an _exact_ algorithm for Card-DSFM running in time $\widetilde{O}\left(T_{mf}(n+R, \mu)\right)$. The authors could mention a statement to this effect, as it seems to be a natural connection.

-Another question I had was regarding the region potentials. Since these correspond to complete graphs, one can very easily sparsify them (by uniformly sampling a few edges from them). I am curious how the performance would compare to the authors' approach.

**Clarity**

The writing is nice in general. I appreciate the authors' sharing the code and data, as well as documentation. I tried to run the code, but it seems that read_data.jl is missing. I also have some minor comments.

-Line 54: “lead” -> “leads”

-Line 148: “solving” -> “solve”

-Line 227: “runtimes” -> “runtime”

-Line 249: “which will faster” -> “which will be faster”

-Line 288: Space missing

-Line 353: “leading to the first approximation algorithms for cardinality-based DSFM” -> “leading to the first approximation algorithms specifically for cardinality-based DSFM”

-Multiple places: “DFSM” -> “DSFM”

**Significance**

This seems like a result that is significant in practice. As the authors note, cardinality-based functions are common in image segmentation, hypergraph clustering, etc, and this paper improves the runtime considerably in the practical examples tested, although for conclusive statements to be made, more experiments would need to be done.

**Time Spent Reviewing:**

3

---

> ### Author Response · Authors · 2021-08-10
> **Reviewer response**
>
> First and foremost, thank you for this very detailed and careful review of our work. We will focus on addressing your concerns regarding our theoretical comparison in Section 4.
>
> Regarding the discussion in lines 214-220, we agree that this paragraph should be changed and re-worded. Allow us to clarify what we mean here. First of all, we do believe that having approximation algorithms (i.e. algorithms with a priori multiplicative approximation guarantees) for CardDSFM is useful. We don't believe there is any disagreement about this point. Beyond this, our main goal in this paragraph is simply to highlight why additive and multiplicative approximations are not directly comparable. Our intent is certainly not to argue that one is always inherently better than another. For example, we included the sentence about logarithmic runtime improvements not to argue that additive approximations are worse, but to make sure it is clear to readers why our approximation algorithms are different in nature from previous additive approximation methods for DSFM. This sentence is also intended to highlight why in our theoretical runtime comparisons we just assume all of these methods are run until optimality (since we ignore log factors anyways). We will update this paragraph to better clarify the point that our approximations are different in nature, but that this is not something that makes them inherently better.
>
> Regarding theoretical runtime comparisons, we will update Section 4 to reiterate here as well that other DSFM algorithms can solve a more general problem. This is of course a benefit to these methods.
>
> Regarding the observation that "certain runtimes from previous work can be improved in the case of cardinality functions", note that in our runtime comparison we do already report improved runtimes for other methods that are obtained when we assume cardinality-based components. We have made a careful search of the literature and have endeavored to use the very best theoretical guarantees that can be derived from existing results. As we highlighted, the best runtime guarantees depend on certain oracle functions, which have faster runtimes in the case of cardinality-based functions. However, our method maintains the same theoretical runtime advantages even if we generously assume the best case $O(|e| \log |e|)$ oracle runtimes for other methods.
>
> Regarding your specific example for the runtime of the method from [4], unless we are misunderstanding your point we believe there is a slight error in your analysis. We are aware that for this method a slightly weaker oracle function suffices. We will update the text to add this point. However, even if the oracle for e only requires sorting, sorting takes $\theta_e = O(|e| \log |e|)$ time, so the expression $\sum_e |e|^2 \theta_e$ does not turn into $\sum_e |e|^2$ but rather $O(\sum_e |e|^3 \log |e|)$. Either way, this is somewhat moot. Even if this were not the case and this term were just $O(\sum_e |e|^2)$, SparseCard would still have significantly faster theoretical runtimes except in extreme cases where all support sets e are constant size. As we noted in the text, constant size e is the regime where the runtime of [4] dominates.
>
> It is an interesting open question whether new analysis could yield even better runtimes for other methods. Indeed, we would be happy to see our work spark an interest in other improved methods and theoretical runtime guarantees specifically for Card-DSFM. However, after a careful search of the literature, we believe that Table 1 and our runtime analysis does provide an accurate picture of the best theoretical guarantees for Card-DSFM that currently exist.
>
> Thank you for the interesting suggestion about combining our graph reduction techniques with the approach in [4], this sounds like an interesting direction for further consideration! Regarding your question about sparsifying complete graphs with uniform sampling: this would also be an interesting question to pursue further. We have separately considered some theoretical regimes where our sparsifiers will be asymptotically better than just sampling edges from a clique, though we have not yet experimented with this in practice. We would be happy to elaborate on these open research directions in the next version of the paper.
>
> Finally, our apologies for not including read_data.jl in the supplement. We will be sure to include this when we release all of our code.

---

### Official Review · Reviewer_YQq9 · 2021-07-16

**Rating:** 6
**Confidence:** 3

**Summary:**

This paper proposes a fast approximate method for minimizing decomposable submodular function where each component is cardinality-based. The runtime is improved by exploiting sparse piecewise linear approximation to concave functions at integer points, which is combined with graph reduction to convert the original minimization problem as an s-t cut problem on a sparse graph. Empirical results show that the mehtod can be faster than state-of-the-art numerical methods. Moreover, the approximation ratios are generally much smaller than the worst-case approximation guarantee.

**Limitations And Societal Impact:**

I cannot think of any potential negative societal impact based on this work.

**Main Review:**

Decomposable submodular function minimization has many applications in machine learning. This paper considers a specific setting where the components are cardinality-based submodular functions. While the results obtained in this paper are interesting, they are not very surprising.

Overall, the paper is very well written and nicely structured. Theoretical analyses appear to be adequate. The proposed method can be a nice tool to have. However, the experiments do not seem to provide a strong evidence for the significance of this work. In particular, the running time of existing numerical methods does not seem to be a limiting factor that prevents the use of cardinality-based submodular functions in the two applications presented in the experiments. For example, for the hypergraph clustering case, why can't ACDM be applied? Are there hypergraph clustering experiments where the proposed method is compared with existing numerical methods?

**Time Spent Reviewing:**

4

---

> ### Author Response · Authors · 2021-08-10
> **Reviewer response**
>
> Thanks for your summary and comments on the paper--in our response we'll focus on addressing your concerns about experiments.
>
> First, to reiterate our main contribution: we have demonstrated in both theory and practice that our method comes with runtime advantages over existing DSFM algorithms in the case of cardinality-based components. These runtime advantages are a beneficial in many different applications, whether competing methods completely fail or are just slower. There are indeed cases where absolute runtimes for competing methods are also fairly fast at solving one DSFM problem, even if not as fast. It is worth noting that in many cases one wishes to solve a large number of individual DSFM problems for a downstream application. In these settings, having a slight runtime improvement for solving a single instance of DSFM can lead a very significant runtime savings overall. For example, in localized clustering, it is common to find many clusters by running the same method for different seed sets and different parameter settings. For our own hypergraph clustering experiments, we have run the HyperLocal algorithm with 10 different settings (counting different locality parameters and hyperedge cut functions) on 45 different clusters, with 10 different seed sets for each cluster, for a total of 4500 runs of HyperLocal. Each run of HyperLocal involves solving a sequence of DSFM problems (not just one). Thus, a small runtime improvement for solving a single DSFM problem would be extremely significant here.
>
> To address your specific questions:
>
> "For example, for the hypergraph clustering case, why can't ACDM be applied?"
>
> If desired, ACDM could also be incorporated as a subroutine into hypergraph clustering methods like HyperLocal. However, our new theoretical runtime results and benchmark image segmentation experiments indicate that SparseCard is an even better tool for this task. As highlighted above, even a minor runtime improvement for solving a single DSFM problem would lead to a huge runtime savings when solving many clustering problems. Furthermore, SparseCard is more immediately applicable for improving existing hypergraph clustering methods that already rely on exact graph reduction techniques. Thus, rather than providing additional comparisons against ACDM, we focus in these experiments on demonstrating how our sparse reduction techniques lead to improvements over exact graph reduction techniques, which are ubiquitous in hypergraph clustering.
>
>
> "Are there hypergraph clustering experiments where the proposed method is compared with existing numerical methods?"
>
> If "existing numerical methods" means applying methods like ACDM as subroutines instead of SparseCard, our answer is given above. If you are asking why we have not included comparisons between HyperLocal+SparseCard and other clustering techniques that are not based on mincuts, we'll address that here as well. In short, our goal in these experiments is to show that sparse reduction techniques can significantly improve the runtime for existing clustering algorithms that rely on exact graph reductions. Understanding the tradeoff between different types of methods (e.g., flow-based methods vs. diffusion-based methods vs. other heuristics) is a separate question, and these types of comparisons can be found in experiments from recent work (see, e.g., [32] and [42]). In the original work of Veldt et al. [42], HyperLocal was shown to outperform several other competing baselines. Here we have demonstrated that we can make HyperLocal much faster with little to no change in cluster quality. In later work by Liu et al. [32], diffusion-based methods for hypergraph clustering were shown to outperform many other methods (including HyperLocal) when limited seed set information is available. These diffusion-based methods also rely on exact graph reduction techniques, and therefore our sparse graph reduction techniques can provide the same type of runtime benefit for this method.

---

### Official Review · Reviewer_dwqT · 2021-07-23

**Rating:** 7
**Confidence:** 2

**Summary:**

The paper tackles the problem of cardinality-based decomposable submodular function minimization (Card-DSFM). DSFMs have been a popular topic of research, and the authors claim that the case when every component function is a cardinality based concave function is an important one in practical applications. For this subproblem, they derive the first approximation algorithms via reduction to sparse graph cuts.

Given a concave cardinality function, the aim is to model it via gadgets (directed graphs on which we will solve a max-flow instance). The authors show that the problem of finding such gadgets is equivalent to finding piecewise linear functions which can approximate the original function with minimum number of linear components. This problem is solved using a greedy method, which gives a better bound on the number of pieces than existing methods.

Once these are optimally computed, they are converted into equivalent gadgets. Once all gadgets are combined into a graph, a max-flow instance is run. The obtained minimum cut also gives a solution to the underlying Card-DSFM.

**Limitations And Societal Impact:**

The authors make a genuine effort into analyzing their limitations, and also talk in some detail about possible negative societal effects of their work on impacts of the paper. There is a clear mention of how it only solves a particular class of problems, as compared to existing methods.

**Main Review:**

The results in this paper apply to a wide variety of practical problems, and formulating theoretical bounds for this problem as an approximation algorithm is a valuable contribution.
Theoretical details are rigorously proved and much effort is gone into formalizing definitions and theorems for this problem. As the paper shows, the results outdo previous baseline results theoretically, even in the case of exact computation. These theoretical contributions, by themselves, are valuable and warrant acceptance in my opinion.

The experiments are also strong, and are able to showcase the strength of the method, in terms of getting better approximations in faster time as opposed to competing methods. The paper also showcases the effectiveness in using the proposed technique as an approximate reduction to graphs for existing methods, and shows improvements in runtimes.

The writing overall is of good quality. The authors do a good job of giving intuition about the technical details in words - however, this can still be improved upon in Section 3.


**Time Spent Reviewing:**

4

---

> ### Author Response · Authors · 2021-08-10
> **Reviewer response**
>
> Thank you for your summary and comments on the paper!

---

### Decision · Program_Chairs · 2021-09-27

**Decision:**

Accept (Poster)

**Comment:**

The reviewers appreciate the problem and the technical contribution of the paper.